# Metric—Phase Fields: Decoupling Distance and Sign for Thin-Structure Reconstruction from Unoriented Point Clouds

Jiayi Kong [1]  Xuhui Chen [2,3]  Chen Zong [4]  Fei Hou [2,3]  Junhui Hou [5]  Wenping Wang [6]  Ying He [1]

## Abstract

Neural Signed Distance Functions (SDFs) excel at reconstructing watertight manifolds but fail on thin structures and open boundaries due to strict inside–outside constraints. Conversely, Unsigned Distance Fields (UDFs) accommodate general geometries but suffer from gradient singularities at the zero-level set, hindering optimization and extraction. We introduce Metric—Phase Fields (MPFs), a decoupled implicit representation that separates metric proximity from topological phase. Given an unoriented point cloud, MPFs learn (i) an unsigned metric field $r$ and (ii) a smooth phase field $\theta$, for which we derive a bounded phase indicator $P = \tanh(\beta\theta)$ that provides soft inside–outside cues where they are meaningful. We couple the two fields via a gated-metric formulation with a residual phase injection to obtain a signed implicit function with stable near-surface gradients. The phase coefficient $\beta$ is learnable, allowing MPFs to adaptively control the sharpness of the phase transition and the degree of saturation of the soft sign indicator. Experiments on both synthetic and scanned thin-shell and thin-plate shapes demonstrate that MPFs preserve thin and layered structures more faithfully than recent SDF-based methods, while also enabling more robust training and more reliable surface extraction than UDF-based approaches. Check out MPFs-GitHub for source code and test models.

[1]S-Lab, Nanyang Technological University, Singapore [2]Key Laboratory of System Software (CAS), Institute of Software, Chinese Academy of Sciences, China [3]University of Chinese Academy of Sciences, China [4]School of Mathematics, Nanjing University of Aeronautics and Astronautics, China [5]Department of Computer Science, City University of Hong Kong, Hong Kong SAR, China [6]Department of Computer Science and Engineering, Texas A&M University, USA. Correspondence to: Ying He <yhe@ntu.edu.sg>.

*Proceedings of the 43rd International Conference on Machine Learning*, Seoul, South Korea. PMLR 306, 2026. Copyright 2026 by the author(s).

## 1. Introduction

Reconstructing high-fidelity 3D surfaces from raw, unoriented point clouds is a fundamental problem in geometry processing and 3D vision, with applications in virtual reality, robotics, computer-aided design, and digital content creation. Neural implicit representations have recently emerged as a dominant paradigm: by learning a continuous function over $\mathbb{R}^3$, they enable accurate surface reconstruction via iso-surface extraction while providing differentiable geometry for downstream tasks.

Among neural implicit representations, signed distance functions (SDFs) are particularly popular because they explicitly encode inside–outside information and allow stable surface extraction (e.g., via Marching Cubes (Lorensen & Cline, 1987) or Dual Contouring (Ju et al., 2002)) from the zero level set. However, the success of SDF-based reconstruction pipelines relies on a key geometric assumption: the target shape is a watertight manifold that induces a globally consistent inside–outside partition of space. This assumption is frequently violated in practice. Real-world objects and man-made models often contain thin structures (e.g., fins, petals, and sheet-like parts), surfaces with boundary, and non-manifold configurations. Moreover, for such thin structures, scanning pipelines often produce single-layer point clouds, where samples lie on only one side of a thin sheet (thin-plate geometry), rather than on both sides of a thin wall (thin-shell geometry).[2]

Thin and open geometries challenge SDF-based reconstruction in two complementary ways. First, for sheet-like structures and surfaces with boundary, the notions of interior and exterior can be ambiguous: a single surface sheet does not uniquely partition space, making a globally consistent sign difficult (or ill-posed) to define. As a result, SDF-based methods often implicitly close or solidify such structures,

---

[2]We distinguish geometry from sampling. *Thin-shell* refers to thin-walled solids whose boundary contains two nearby sheets (small thickness). *Thin-plate* refers to single-sheet surfaces (often with boundary or attached non-manifoldly), where a consistent inside/outside label may be ill-posed. *Single-layer point clouds* describe a sampling regime in which points are observed on only one side of a locally thin structure, regardless of whether the underlying geometry is thin-shell or thin-plate.

leading to artifacts such as thickness inflation, spurious closures, or missing sheets. Second, even for watertight shapes, thin shells contain multiple nearby surface layers separated by small gaps. In this setting, learning a single scalar field that must simultaneously capture accurate distances and a stable sign becomes ill-conditioned. When the network over-smooths or predicts inconsistent signs, it may merge layers that should remain separate, fill in thin cavities, or blur fine-scale details.

Unsigned distance fields (UDFs) provide a different perspective. By eliminating the need for a global sign, UDFs naturally accommodate surfaces with boundary and thin multi-layered structures. However, UDFs introduce their own challenges for neural learning and surface extraction. Because UDFs lack sign information for localizing the zero-level set, the extracted meshes often remain suboptimal even when utilizing additional procedures such as dedicated optimization (Hou et al., 2023) or auxiliary information (e.g., gradients (Guillard et al., 2022; Ren et al., 2023)).

We propose Metric–Phase Fields (MPFs), a decoupled implicit representation designed to combine the complementary strengths of SDFs and UDFs. The core idea is to separate metric proximity from sign/phase information: given an unoriented point cloud, MPFs learn (i) an unsigned metric field $r(\mathbf{x})$ that focuses on accurate proximity (UDF-like), and (ii) a smooth phase field $\theta(\mathbf{x})$ that encodes a soft inside–outside preference where such a notion is meaningful. From $\theta$, we derive a bounded phase indicator $P(\mathbf{x}) = \tanh(\beta\,\theta(\mathbf{x}))$, where the learnable coefficient $\beta$ adaptively controls the sharpness of the phase transition and the saturation of the soft sign cue. We then couple the two fields via a gated-metric formulation with residual phase injection, producing a signed implicit function with stable near-surface gradients. This design enables reliable learning of the sign transition without compromising metric accuracy. We analyze the theoretical properties of MPFs and introduce a set of loss terms for learning them directly from unoriented point clouds. Experiments on both synthetic and scanned thin-shell and thin-plate shapes demonstrate that MPFs preserve thin and layered structures more faithfully than recent SDF-based methods, while also enabling more robust training and more reliable surface extraction than UDF-based approaches.

## 2. Related Work

Surface reconstruction from point clouds has been studied for decades, spanning classical computational geometry, variational implicit methods, and, more recently, deep learning. Due to space constraints, we focus on the most relevant works on implicit approaches (both classical and neural) and refer the reader to recent surveys (Berger et al., 2017; Huang et al., 2024) for a broader overview. At a high level,

implicit reconstruction methods can be grouped into signed representations, which rely on an inside–outside partition, and unsigned representations, which avoid global sign but often require additional machinery for surface extraction.

### 2.1. Signed Representations

Early work approximated a signed distance field using local tangent-plane estimates at closest points (Hoppe et al., 1992). Radial basis function (RBF) approaches (Carr et al., 2001; Huang et al., 2019) fit an oriented point set with a globally smooth implicit function. The Poisson Surface Reconstruction (PSR) family, including PSR (Kazhdan et al., 2006), sPSR (Kazhdan & Hoppe, 2013), and iPSR (Hou et al., 2022), is popular due to their efficiency and effectiveness in capturing fine geometric details. A complementary line of work derives sign from generalized winding number (GWN) (Jacobson et al., 2013). Recent GWN-based methods, such as GCNO (Xu et al., 2023), BIM (Liu et al., 2024), WNNC (Lin et al., 2024), and DWG (Liu et al., 2025), have demonstrated strong robustness for watertight manifold surfaces from unoriented or defective point sets.

Neural signed representations, in particular neural SDFs, have also been explored extensively due to their expressive power and the availability of effective geometric regularizers, such as the first-order Eikonal constraint and higher-order Hessian-based priors (e.g., singular Hessian constraints). These priors encourage distance-like behavior and stable gradients, making SDFs a popular choice for learning-based surface reconstruction and downstream geometric optimization, as demonstrated in representative methods, such as Neural-Pull (Ma et al., 2021), DiGS (Ben-Shabat et al., 2022), StEik (Yang et al., 2023), NSH (Wang et al., 2023a), NeuralGF (Li et al., 2023), and NeuralCADRecon (Dong et al., 2024), among others. Points2Surf (Erler et al., 2020) further explores joint learning of signed distance and surface reconstruction from point clouds by decoding both sign and distance information from a shared feature representation. However, coupling sign and metric prediction within a unified latent space may lead to inconsistencies near the surface, where precise geometric alignment is required for accurate reconstruction.

Another widely used neural implicit representation is occupancy fields (Mescheder et al., 2019), which model a shape as a continuous classifier $o(\mathbf{x}) \in [0, 1]$ indicating whether $\mathbf{x}$ lies inside the object. Surfaces are then extracted as an iso-contour $o(\mathbf{x}) = 0.5$ using standard iso-surfacing. ConvONet (Peng et al., 2020) improves reconstruction quality by combining convolutional encoders with implicit occupancy decoders. POCO (Boulch & Marlet, 2022) and ALTO (Wang et al., 2023b) further enhance local geometric detail via point-based convolutions and adaptive latent representations.

Despite their success on watertight manifolds, signed representations are fundamentally challenged by surfaces with boundary, thin sheets, and non-manifold configurations, where a globally consistent inside–outside labeling can be ambiguous or ill-posed.

## 2.2. Unsigned Representations

Unlike occupancy or signed representations, unsigned distance fields discard sign information and therefore provide a more general formulation for modeling open surfaces, non-manifold structures, and geometries without a globally consistent inside–outside partition. NDF (Chibane et al., 2020) learns a UDF from input point clouds using an MLP. GeoUDF (Ren et al., 2023) improves UDF quality and gradient estimation through geometric guidance, but incurs substantial memory overhead. Inspired by Neural-Pull (Ma et al., 2021), CAP-UDF (Zhou et al., 2024) introduces field-consistency losses, while LevelSetUDF (Zhou et al., 2023) enforces continuity through level-set projection. More specialized formulations, such as UODF (Lu et al., 2024) and DUDF (Fainstein et al., 2024), introduce orthogonal or hyperbolic distance scalings to improve representational flexibility. LoSF-UDF (Hu et al., 2025) leverages local shape functions for UDF learning, and DEUDF (Xu et al., 2025) incorporates normal supervision to enhance detail preservation. Beyond neural UDF learning, VAD (Kong et al., 2025) proposes a lightweight, network-free approach that computes UDFs from unoriented point clouds by aligning bidirectional normals with Voronoi-based criteria, diffusing the resulting normal field, and integrating it into a UDF. Despite these advances, extracting high-quality surfaces from UDFs remains challenging, often leading to holes (Zhou et al., 2024; Ren et al., 2023; Guillard et al., 2022), double-layer surfaces (Hou et al., 2023; Chen et al., 2025b), or unwanted non-manifold artifacts (Zhang et al., 2023; Chen et al., 2025a).

## 3. Metric–Phase Fields

Let $\mathbf{x} \in \mathbb{R}^3$ denote a query location. Let $\mathcal{S} \subset \mathbb{R}^3$ be the (unknown) target surface and let $\mathcal{X} = \{\mathbf{p}_i\}$ be the input unoriented point cloud sampled from $\mathcal{S}$. We learn a metric field $r(\mathbf{x})$, a phase field $\theta(\mathbf{x})$, and use their combination $\phi(\mathbf{x})$ as the signed implicit function.

## 3.1. Motivations

Conventional SDF-based reconstruction represents a surface using a single scalar field that is expected to encode both (i) metric proximity to the surface and (ii) a globally consistent inside–outside sign. This coupling is well-suited to watertight, two-sided manifold surfaces, where a global inside–outside partition exists and the signed distance exhibits stable, well-behaved gradients near the surface. How-

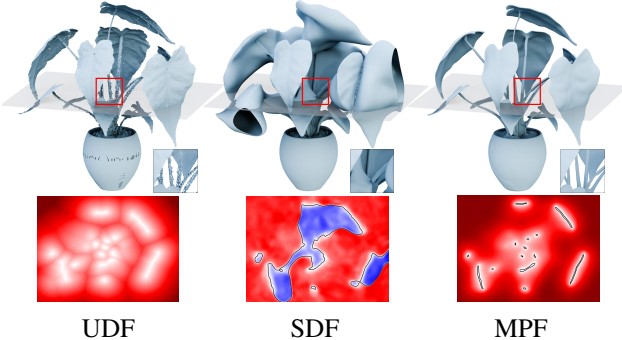

UDF       SDF       MPF

*Figure 1.* Challenges of SDFs and UDFs in surface reconstruction. When the inside–outside sign is ambiguous (e.g., in thin plates, boundary regions, or non-manifold configurations), learning a single SDF becomes ill-conditioned and may introduce artifacts such as inflated thickness. In contrast, UDFs often suffer from optimization difficulties and unstable zero level set extraction due to non-differentiability at $u = 0$ and the lack of sign information. Black lines in the 2D slices indicate the zero-level set.

ever, many practical inputs violate these assumptions. For shapes containing open boundaries, non-manifold configurations, or thin-plate (sheet-like) structures, the notion of a global sign becomes ambiguous or ill-posed. In such cases, forcing a single field to simultaneously satisfy distance-like behavior and sign consistency often leads to reconstruction artifacts (Figure 1).

## 3.2. Decomposition of Metric and Phase

To address this issue, we decouple geometric proximity and sign/phase information into two fields with complementary roles: a *Metric Field* $r(\mathbf{x})$ that captures unsigned proximity, and a *Phase Field* $\theta(\mathbf{x})$ that provides a soft interior/exterior preference where such a notion is meaningful.

**Metric Field.** The metric field $r(\mathbf{x})$ is trained to behave as an unsigned, UDF-like distance and is regularized using an Eikonal constraint on off-surface samples:

$$\|\nabla r(\mathbf{x})\|_2 = 1. \tag{1}$$

We apply this constraint away from the interface (i.e., excluding points with $r(\mathbf{x}) \approx 0$), since a true unsigned distance field is generally non-differentiable at its zero level set due to cusp-like behavior.

To ensure a valid unsigned metric, we enforce non-negativity by construction. Specifically, we parameterize $r$ using a softplus activation function, which guarantees $r(\mathbf{x}) \geq 0$ and maintains smoothness of the metric branch, while still allowing $r(\mathbf{x})$ to approach zero on surface samples under the data constraints.

**Phase field.** The phase field $\theta(\mathbf{x})$ acts as a continuous latent phase descriptor. Unlike discrete occupancy/sign

classification, we derive a soft phase indicator $P(\mathbf{x})$ through a hyperbolic tangent:

$$P(\mathbf{x}) = \tanh\big(\beta\,\theta(\mathbf{x})\big), \qquad (2)$$

where $P(\mathbf{x}) \in (-1, 1)$ provides a soft phase label (inside/outside preference) and $\beta > 0$ is a learnable scaling factor. Modeling $\theta(\mathbf{x})$ as a continuous field avoids the instability associated with discrete sign switching.

This separation allows each component to be trained with objectives aligned to its role, while their combination yields a signed implicit function with improved robustness on challenging geometries.

### 3.3. Residual Gradient Injection

We combine the metric and phase components using a gated-metric formulation with a residual phase injection:

$$\phi(\mathbf{x}) = r(\mathbf{x})\,P(\mathbf{x}) + \theta(\mathbf{x}), \qquad (3)$$

where $P(\mathbf{x})$ gates the metric term and $\theta(\mathbf{x})$ serves as an additive residual that stabilizes optimization near the interface.

In our formulation, the reconstructed surface is obtained as the zero level set $\{\mathbf{x} : \phi(\mathbf{x}) = 0\}$; in particular, we enforce $\phi(\mathbf{p}) = 0$ on input samples $\mathbf{p} \in \mathcal{X}$. In the following, we show that the zero level set of $\phi$ coincides with the zero level set of $\theta$, and they share the same gradients at constrained samples.

**Proposition 3.1** (Sign consistency). *Let $P(\mathbf{x}) = \tanh(\beta\,\theta(\mathbf{x}))$ with $\beta > 0$ and $\phi = rP + \theta$. Then for any $\mathbf{x}$ with $\theta(\mathbf{x}) \neq 0$,*

$$\mathrm{sign}\big(\phi(\mathbf{x})\big) = \mathrm{sign}\big(\theta(\mathbf{x})\big), \qquad (4)$$

*and moreover*

$$\phi(\mathbf{x}) = 0 \quad \Longleftrightarrow \quad \theta(\mathbf{x}) = 0. \qquad (5)$$

*Proof.* Since $\tanh(\cdot)$ is odd and strictly increasing and $\beta > 0$, we have $\mathrm{sign}(P(\mathbf{x})) = \mathrm{sign}(\theta(\mathbf{x}))$ and $P(\mathbf{x}) = 0 \Leftrightarrow \theta(\mathbf{x}) = 0$. Because $r(\mathbf{x}) \geq 0$ (by our construction), the product $r(\mathbf{x})P(\mathbf{x})$ has the same sign as $P(\mathbf{x})$ (or is zero). Thus, whenever $\theta(\mathbf{x}) \neq 0$ (equivalently $P(\mathbf{x}) \neq 0$), the two terms $\theta(\mathbf{x})$ and $r(\mathbf{x})P(\mathbf{x})$ share the same sign, so $\phi(\mathbf{x}) = \theta(\mathbf{x}) + r(\mathbf{x})P(\mathbf{x})$ cannot change sign relative to $\theta(\mathbf{x})$. Finally, $\phi(\mathbf{x}) = 0$ implies $\theta(\mathbf{x}) = 0$, and the reverse direction holds because $\theta(\mathbf{x}) = 0$ gives $P(\mathbf{x}) = 0$ and hence $\phi(\mathbf{x}) = 0$. $\qquad\square$

Proposition 3.1 implies that under $r(\mathbf{x}) \geq 0$ and $\beta > 0$, the zero level set of $\phi$ coincides with that of the phase field, i.e., $\{\phi = 0\} = \{\theta = 0\}$. Importantly, $\phi(\mathbf{x}) = 0$ does *not* imply $r(\mathbf{x}) = 0$: when $\theta(\mathbf{x}) = 0$ we have

$P(\mathbf{x}) = \tanh(\beta\theta(\mathbf{x})) = 0$, and thus $\phi(\mathbf{x}) = 0$ regardless of the value of $r(\mathbf{x})$. For this reason, during training we explicitly enforce *both* constraints on samples $\mathbf{p} \in \mathcal{X}$ by penalizing deviations of $\phi(\mathbf{p})$ and $r(\mathbf{p})$ from zero (see $\mathcal{L}_{\mathrm{zero}}$ in Section 3.4).

**Proposition 3.2** (Gradient agreement at constrained samples). *For any point $\mathbf{p}$ satisfying $r(\mathbf{p}) = 0$ and $\phi(\mathbf{p}) = 0$ (in particular, for training samples $\mathbf{p} \in \mathcal{X}$ where both constraints are enforced), we have*

$$\nabla\phi(\mathbf{p}) = \nabla\theta(\mathbf{p}). \qquad (6)$$

*Proof.* Differentiating (3) gives

$$\nabla\phi(\mathbf{x}) = P(\mathbf{x})\,\nabla r(\mathbf{x}) + r(\mathbf{x})\,\nabla P(\mathbf{x}) + \nabla\theta(\mathbf{x})$$
$$= P(\mathbf{x})\nabla r(\mathbf{x}) + \big(1 + r(\mathbf{x})\beta\,\mathrm{sech}^2\,(\beta\theta(\mathbf{x}))\big)\,\nabla\theta(\mathbf{x}),$$

where we used $\nabla P(\mathbf{x}) = \beta\,\mathrm{sech}^2(\beta\,\theta(\mathbf{x}))\,\nabla\theta(\mathbf{x})$. At a constrained sample $\mathbf{p}$, $r(\mathbf{p}) = 0$ and $\phi(\mathbf{p}) = 0$ imply $\theta(\mathbf{p}) = 0$, hence $P(\mathbf{p}) = 0$. Substituting into the above equation yields $\nabla\phi(\mathbf{p}) = \nabla\theta(\mathbf{p})$. $\qquad\square$

Proposition 3.2 shows that at surface samples (i.e., samples on the observed surface where loss terms drive $r(\mathbf{p})$ and $\theta(\mathbf{p})$ towards zero), the composite field $\phi$ and the phase field $\theta$ agree to first order at convergence. Specifically, for a regular implicit surface point (i.e., $\nabla\phi(\mathbf{p}) \neq \mathbf{0}$), the surface normal is given by $\mathbf{n}_\phi(\mathbf{p}) = \nabla\phi(\mathbf{p})/\|\nabla\phi(\mathbf{p})\|_2$. Therefore, at constrained samples, the level sets $\widehat{\mathcal{S}}_\phi = \{\mathbf{x} : \phi(\mathbf{x}) = 0\}$ and $\widehat{\mathcal{S}}_\theta = \{\mathbf{x} : \theta(\mathbf{x}) = 0\}$ share the same normal (and thus the same tangent plane) at $\mathbf{p}$:

$$\mathbf{n}_\phi(\mathbf{p}) = \frac{\nabla\phi(\mathbf{p})}{\|\nabla\phi(\mathbf{p})\|_2} = \frac{\nabla\theta(\mathbf{p})}{\|\nabla\theta(\mathbf{p})\|_2} = \mathbf{n}_\theta(\mathbf{p}).$$

Equivalently, the gated metric term is *first-order inactive* at constrained samples. Indeed, since $r(\mathbf{p}) = 0$ and $P(\mathbf{p}) = 0$, we have $(rP)(\mathbf{p}) = 0$ and

$$\nabla(rP)(\mathbf{p}) = P(\mathbf{p})\,\nabla r(\mathbf{p}) + r(\mathbf{p})\,\nabla P(\mathbf{p}) = \mathbf{0}.$$

Thus the gated metric does not alter the first-order interface geometry at these points; instead, it primarily shapes $\phi$ away from the interface by injecting metric proximity with a soft sign. This observation explains how MPFs retain stable, well-defined interface normals (as in SDFs on smooth watertight surfaces), while still leveraging a UDF-like metric branch $r$ that remains meaningful in sign-ambiguous regions.

### 3.4. Constraints and Regularization

To learn MPFs from unoriented points, we design the following loss terms.

**Surface Data constraints.** To ensure the implicit surface interpolates the input point cloud $\mathcal{X} = \{\mathbf{p}_i\}$, we impose zero-level consistency on both the metric and composite fields:

$$\mathcal{L}_{\text{zero}} = \sum_{\mathbf{p} \in \mathcal{X}} \left(r(\mathbf{p}) + |\phi(\mathbf{p})|\right). \tag{7}$$

While $\phi(\mathbf{p}) = 0$ enforces surface consistency of the composite field, the additional constraint $r(\mathbf{p}) = 0$ prevents metric drift and keeps the distance component geometrically anchored.

**Gradient alignment (near-surface).** We encourage the metric and phase fields to share consistent normal directions by aligning their *gradient directions* in a near-surface band. This links phase transitions to geometric distance while preserving the functional separation between distance magnitude ($r$) and phase ($\theta$). For surfaces with boundary, this term is particularly important: standard SDF-based objectives often bias the solution toward a watertight inside–outside partition, which can introduce auxiliary zero level sets in free space. For closed (watertight) surfaces, where a global sign is well-defined, the same loss serves as a stabilizing coupling regularizer that reduces local sign inconsistencies and helps $\theta$ track the geometric normals implied by $r$ under sparse/noisy sampling. Formally, we use

$$\mathcal{L}_{\text{align}} = \mathbb{E}_{\mathbf{x}} \Big[ w\left(r(\mathbf{x})\right) \Big(1 - \Big| \frac{\nabla r(\mathbf{x})}{\|\nabla r(\mathbf{x})\|_2 + \varepsilon} \\ \cdot \frac{\nabla \theta(\mathbf{x})}{\|\nabla \theta(\mathbf{x})\|_2 + \varepsilon} \Big| \Big) \Big], \tag{8}$$

where $w(r) = \exp(-\alpha r)$ concentrates the constraint to a narrow band around the surface, and the small constant $\varepsilon$ prevents numerical instability caused by vanishing gradients. The absolute inner product makes the constraint invariant to orientation, which is desirable for unoriented point clouds.

**Metric Eikonal constraint (near-surface).** For the metric component $r(\mathbf{x})$, we encourage it to approximate a Euclidean unsigned distance field. A true UDF is generally not differentiable at the zero level set (the interface), due to cusp-like behavior analogous to an absolute value function. Therefore, we do not enforce the Eikonal constraint exactly on surface samples. Instead, we impose it on a narrow band of off-surface samples:

$$\mathcal{N}_\delta := \{\mathbf{x} \in \Omega : 0 < r(\mathbf{x}) < \delta\}, \tag{9}$$

$$\mathcal{L}_{\text{eik}}^r = \mathbb{E}_{\mathbf{x} \in \mathcal{N}_\delta} \left[|\|\nabla r(\mathbf{x})\|_2 - 1|\right], \tag{10}$$

where $\Omega$ denotes the spatial domain of interest. This term encourages $r(\mathbf{x})$ to grow linearly away from the geometry while avoiding the non-differentiable interface.

**Topological Eikonal constraint (on-sample interface).** In contrast to $r$, the composite field $\phi$ is designed to behave like a signed implicit function near the boundary. On surface samples $\mathbf{p} \in \mathcal{X}$, we explicitly enforce $r(\mathbf{p}) = 0$ and $\phi(\mathbf{p}) = 0$, which implies $\theta(\mathbf{p}) = 0$ and yields $\nabla\phi(\mathbf{p}) = \nabla\theta(\mathbf{p})$ (Proposition 3.2). We therefore enforce a unit-norm constraint on the sampled interface:

$$\mathcal{L}_{\text{eik}}^\phi = \mathbb{E}_{\mathbf{p} \in \mathcal{X}} \left[|\|\nabla\phi(\mathbf{p})\|_2 - 1|\right] + \mathbb{E}_{\mathbf{p} \in \mathcal{X}} \left[|\|\nabla\theta(\mathbf{p})\|_2 - 1|\right]. \tag{11}$$

**Laplacian regularization.** To suppress spurious oscillations and encourage simple field behavior away from the reconstructed surface, we apply a second-order smoothness prior to the composite field $\phi$. Similar to DiGS (Ben-Shabat et al., 2022), we regularize $\phi$ to be approximately harmonic in free space by penalizing the Laplacian magnitude:

$$\mathcal{L}_{\text{lap}} = \mathbb{E}_{\mathbf{x} \in \Omega} \left[\left|\Delta\phi(\mathbf{x})\right|\right], \tag{12}$$

where $\Delta\phi = \nabla \cdot \nabla\phi$ denotes the Laplacian. In practice, the Laplacian term is evaluated using sparse samples in the ambient space, which regularizes the global field behavior while minimizing interference with surface accuracy and thin structures.

**Phase saturation constraint.** While the phase field $\theta(\mathbf{x})$ is continuous, the derived soft phase indicator $P(\mathbf{x}) = \tanh(\beta\,\theta(\mathbf{x}))$ should saturate toward $\{-1, +1\}$ in regions away from the interface. To avoid ambiguous states where $P(\mathbf{x}) \approx 0$ in free space (which can thicken or blur the transition), we impose:

$$\mathcal{L}_{\text{phase}} = \mathbb{E}_{\mathbf{x} \in \Omega \setminus \mathcal{N}_\delta} \left[\left(1 - |P(\mathbf{x})|\right)^2\right]. \tag{13}$$

**Far-field repulsion.** To suppress spurious zero-level sets away from the observed surface, we impose a repulsion constraint on both the metric and composite fields in regions far from the interface. Specifically, we penalize *small* field magnitudes using an exponential barrier:

$$\mathcal{L}_{\text{far}} = \mathbb{E}_{\mathbf{x} \in \Omega \setminus \mathcal{N}_\delta} \left[\exp\left(-\alpha\left(r(\mathbf{x}) + |\phi(\mathbf{x})|\right)\right)\right], \tag{14}$$

where $\Omega$ denotes the sampling domain and $\mathcal{N}_\delta$ is a near-surface band. This loss is large only when both $r(\mathbf{x})$ and $|\phi(\mathbf{x})|$ are close to zero in free space, precisely the situation that would indicate an undesired interface or a spurious sheet far from the data. As $r(\mathbf{x})$ grows and/or $|\phi(\mathbf{x})|$ moves away from zero, the exponential term decays rapidly, so the penalty becomes negligible and does not constrain the field unnecessarily. The parameter $\alpha$ controls the sharpness of the barrier: larger values concentrate the penalty more strongly on near-zero magnitudes. In our experiments, we set $\alpha = 100$ to make the repulsion effective only when $r(\mathbf{x}) + |\phi(\mathbf{x})|$ is very small, which efficiently discourages far-field zero-crossings while keeping the loss inactive elsewhere.

*Figure 2.* Pipeline. Query points sampled from an unoriented point cloud are processed by a SIREN-based network to predict $r(\mathbf{x})$ and $\theta(\mathbf{x})$, which are composed into the MPF $\phi(\mathbf{x})$. The reconstructed mesh is directly extracted using Marching Cubes.

**(Optional) Normal alignment.** For dense and low-noise point clouds, one can estimate reliable *local* (unoriented) normals via PCA, although their global orientation may be inconsistent. When such local normals are available, we optionally impose a normal alignment loss to encourage consistency between the learned implicit field and the input geometry. Specifically, for each input sample $\mathbf{p} \in \mathcal{X}$ with an estimated unit normal $\mathbf{n}(\mathbf{p})$, we encourage the gradient direction of the composite field to align with $\mathbf{n}(\mathbf{p})$. Since PCA normals are unoriented, we use an orientation-invariant objective:

$$\mathcal{L}_{\text{normal}} = \mathbb{E}_{\mathbf{p} \in \mathcal{X}} \left[ 1 - \left| \frac{\nabla \phi(\mathbf{p})}{\|\nabla \phi(\mathbf{p})\|_2} \cdot \mathbf{n}(\mathbf{p}) \right| \right]. \quad (15)$$

This term can improve geometric fidelity when normal estimates are reliable, similar in spirit to the normal/gradient supervision used in DEUDF (Xu et al., 2025).

### 3.5. Network Architecture

Our network is a straightforward design of the two continuous fields over $\mathbb{R}^3$: an unsigned metric field $r(\mathbf{x})$ and a phase field $\theta(\mathbf{x})$. Given a query location $\mathbf{x} \in \mathbb{R}^3$, we first embed it with a shared MLP backbone $f_\psi : \mathbb{R}^3 \to \mathbb{R}^{256}$, and then predict $r(\mathbf{x})$ and $\theta(\mathbf{x})$ using two lightweight heads (Figure 2). We use a fully-connected backbone with sinusoidal activations (SIREN-style) to capture high-frequency geometric details. Concretely, the backbone consists of two linear layers $3 \to 256 \to 256$, each followed by a sine nonlinearity, producing a feature vector $\mathbf{f}(\mathbf{x}) = f_\psi(\mathbf{x}) \in \mathbb{R}^{256}$.

The metric (geometry) head $g_r$ maps $\mathbf{f}(\mathbf{x})$ to a scalar metric prediction. We use a small MLP $256 \to 256 \to 1$ with a sine nonlinearity, followed by a nonnegative activation:

$$r(\mathbf{x}) = \text{softplus}\left(g_r(\mathbf{f}(\mathbf{x}))\right) \geq 0. \quad (16)$$

This guarantees nonnegativity of the metric branch by construction. In our implementation we use a sharp softplus (e.g., slope = 100) so that $r(\mathbf{x})$ can approach zero on surface samples while remaining smooth.

The phase head $g_\theta$ has the same lightweight structure $256 \to 256 \to 1$ (with a sine nonlinearity), but outputs an unbounded scalar:

$$\theta(\mathbf{x}) = g_\theta(\mathbf{f}(\mathbf{x})) \in \mathbb{R}. \quad (17)$$

When local topology is expected to be consistent (e.g., for closed/watertight shapes), we optionally initialize the final bias of the phase head to a small negative value to encourage a stable initial phase separation; otherwise we use standard initialization.

From $\theta(\mathbf{x})$, we derive a bounded phase indicator $P(\mathbf{x}) = \tanh\left(\beta\theta(\mathbf{x})\right)$, where the learnable scalar $\beta > 0$ controls the sharpness of the phase transition. Finally, we form the signed composite field using our gated-metric formulation with residual phase injection $\phi(\mathbf{x}) = r(\mathbf{x})P(\mathbf{x}) + \theta(\mathbf{x})$ and extract the surface as the zero level set $\{\mathbf{x} : \phi(\mathbf{x}) = 0\}$.

We optimize the backbone parameters $\psi$, head parameters, and the scalar $\beta$ by minimizing a weighted combination of the losses introduced in Section 3.4:

$$\min \mathcal{L}_{\text{zero}} + \lambda_{\text{align}}\mathcal{L}_{\text{align}} + \lambda_{\text{eik-r}}\mathcal{L}_{\text{eik}}^r + \lambda_{\text{eik-}\phi}\mathcal{L}_{\text{eik}}^\phi + \lambda_{\text{lap}}\mathcal{L}_{\text{lap}}$$
$$+ \lambda_{\text{far}}\mathcal{L}_{\text{far}} + \lambda_{\text{phase}}\mathcal{L}_{\text{phase}} + \lambda_{\text{normal}}\mathcal{L}_{\text{normal}}, \quad (18)$$

where $\lambda$s balance the contributions of each term. The normal alignment loss $\mathcal{L}_{\text{normal}}$ is optional and is enabled only when reliable local normal estimates are available.

## 4. Relation to SDFs and UDFs

For a smooth, watertight, two-sided surface $\mathcal{S}$, a (local) signed distance function $s(\mathbf{x})$ satisfies $s(\mathbf{x}) = 0$ on $\mathcal{S}$ and $\|\nabla s(\mathbf{x})\|_2 = 1$ wherever it is differentiable in a tubular neighborhood of $\mathcal{S}$ (excluding singularities where the closest point is not unique, e.g., on the medial axis). In contrast, an unsigned distance field $u(\mathbf{x}) = \text{dist}(\mathbf{x}, \mathcal{S})$ is nonnegative and does not encode inside–outside sign. Near a smooth surface, $u$ behaves like an absolute value function along the normal direction, and is therefore generally *non-differentiable* on the interface $u = 0$. For surfaces with boundary, small-offset level sets $\{u = \varepsilon\}$ tend to form a two-sided shell that wraps around boundary edges, often described as a *double-cover* effect.

MPFs are closely related to both signed and unsigned distance representations, but differ in how they allocate responsibilities between *metric* and *sign/phase*. Like an SDF, MPFs provide a signed implicit function suitable for iso-surface extraction, via the composite field $\phi$ (with phase information carried by $\theta$). Unlike SDF regression, where a

single scalar must simultaneously fit distances and maintain global sign consistency, MPFs decouple these objectives: the metric branch $r$ is trained as a UDF-like proximity field, while the phase branch $\theta$ models the sign transition.

Like a UDF, the metric field $r$ remains meaningful for surfaces with boundary, thin sheets, and certain non-manifold regions where a globally consistent sign is ambiguous. Correspondingly, MPFs may inherit the UDF tendency to form a narrow two-sided shell around such structures. At the same time, MPFs mitigate two practical drawbacks of pure UDFs: (i) we enforce the Eikonal constraint for $r$ only on near-surface *off-interface* samples (e.g., $0 < r < \delta$), avoiding the cusp at $r = 0$ and singular sets such as the medial axis; and (ii) the residual phase injection yields stable, well-defined *interface normals* at constrained samples (via $\nabla\phi = \nabla\theta$ when $r = 0$ and $\phi = 0$), enabling SDF-style normal estimation and on-sample unit-gradient constraints.

A key practical difference is iso-surface extraction. UDFs are nonnegative and therefore do not exhibit a sign change across the target surface; consequently, standard MC, which relies on sign changes across grid edges, is not directly applicable to the zero set $\{u = 0\}$ in a robust way. In practice, UDF-based reconstruction often relies on additional extraction procedures, such as dedicated optimization (Hou et al., 2023) or auxiliary gradient predictions (Ren et al., 2023). In contrast, MPFs produce a signed composite field $\phi$, so the target surface can be extracted directly as $\{\phi = 0\}$ using standard MC. For thin plates and sign-ambiguous configurations (e.g., boundary edges and non-manifold junctions), MPFs often form a narrow double-cover shell: $\phi$ takes opposite signs on the two sides of the sheet, separated by a small gap, and therefore crosses zero between them. This built-in sign transition makes the zero crossing well-conditioned for MC, while retaining UDF-like behavior around open structures. Moreover, the residual phase injection provides a direct gradient pathway for the phase branch, yielding more stable near-interface optimization than the cusp-like behavior of a true UDF.

Overall, MPFs can be viewed as combining the *metric robustness* of UDFs with the *signed interface behavior* of SDFs, while avoiding the ill-conditioning that arises when both must be learned by a single field. Table 1 summarizes the key differences. Figure 3 illustrates the complementary failure modes of SDFs and UDFs on a shape that mixes a watertight component with a thin, sign-ambiguous sheet. The SDF branch cleanly separates the cube interior/exterior but tends to distort the sheet region, while the UDF branch represents the sheet naturally but lacks sign information even on the cube. MPF combines the two behaviors by keeping a signed field on the watertight part and falling back to a UDF-like shell around the thin sheet, so that standard MC can still extract the surface directly from $\{\phi = 0\}$. Figure 6

further visualizes the reconstructed surfaces together with the intermediate fields underlying this transition behavior.

## 5. Experiments

**Implementation details.** All experiments are conducted on a single NVIDIA GeForce RTX 4090 GPU with 24 GB of memory. Input coordinates are normalized to the range $[-1, 1]^3$. We construct near-surface tubular band $\mathcal{N}_\delta$ by jittering the input samples. Since the true surface $\mathcal{S}$ is unknown, we approximate the unsigned distance to the surface by the nearest-neighbor distance to the input point set, $d_{\mathcal{X}}(\mathbf{x}) = \min_{\mathbf{p}_i \in \mathcal{X}} \|\mathbf{x} - \mathbf{p}_i\|_2$, computed using a KD-tree, and retain samples satisfying $0 < d_{\mathcal{X}}(\mathbf{x}) < \delta$ as near-surface points. Far-field samples in $\Omega \setminus \mathcal{N}_\delta$ are drawn uniformly from the bounding box $\Omega$ of the input point cloud and rejected if $d_{\mathcal{X}}(\mathbf{x}) < \delta$. Unless otherwise specified, the loss weights $(\lambda_{\text{align}}, \lambda_{\text{eik-}r}, \lambda_{\text{eik-}\phi}, \lambda_{\text{lap}}, \lambda_{\text{phase}}, \lambda_{\text{far}})$ are set to $(0.7, 0.007, 0.007, 0.0004, 0.002, 0.1)$. When reliable local normals can be estimated easily, e.g., via PCA, we set $\lambda_{\text{normal}} = 0.15$; otherwise, we set it to 0. In our implementation, $\beta$ is initialized to $50$ and optimized with a learning rate of $1e{-}4$. As described in the appendix, this setting provides a good practical balance between convergence speed and numerical stability. To stabilize early training, we activate $\mathcal{L}_{\text{align}}$ only after 3,000 iterations.

**Test models.** Since MPFs are designed to combine the strengths of signed and unsigned representations, we evaluate on models that pose complementary challenges to both classes of methods. Specifically, we select 30 models from Thingi10K (Zhou & Jacobson, 2016) and ShapeNet (Chang et al., 2015), together with 5 artist-designed models from Sketchfab. These models contain thin shells, thin plates, open boundaries, and closely spaced surface layers. Such structures are challenging for SDF-based methods because the inside–outside labeling can be ambiguous, ill-posed, or numerically unstable near thin regions. The test set also includes fine geometric details and closely spaced features, which are difficult for UDF-based learning due to the non-differentiability of unsigned distance fields at the zero set and the resulting optimization instability. Together, these examples form a balanced benchmark for evaluating both geometric fidelity and robustness.

**Baselines.** We compare our method against two groups of state-of-the-art neural implicit reconstruction methods. The first group consists of SDF-based methods, including DiGS (Ben-Shabat et al., 2022), NSH (Wang et al., 2023a), StEik (Yang et al., 2023), PG-SDF (Koneputugodage et al., 2024), and I-filtering (Li et al., 2025). These methods learn signed fields and are primarily designed for watertight, two-sided manifold surfaces, where a globally consistent inside–outside labeling is well defined. The second group consists

*Table 1.* Comparison of implicit representations. MPFs decouple metric proximity ($r$) from phase/sign ($\theta$) and combine them via the composite field $\phi$.

| | SDF | UDF | Ours (metric–phase) |
|---|---|---|---|
| Sign/inside–outside | ✓ | × | ✓ (via $\theta$ / $\phi$) |
| Metric proximity | ✓ | ✓ | ✓ (via $r$) |
| Eikonal constraint | near-surface band $-\delta < s < \delta$; avoid medial axis | near-surface band $0 < u < \delta$; avoid medial axis | $r$: near-surface band $0 < r < \delta$; avoid medial axis; $\phi$ (or $\theta$): on-sample interface |
| Differentiable on the interface | yes | no at $u = 0$ (cusp) | yes at constrained samples (via $\theta$) |
| Surfaces with boundary/thin sheets | sign ill-posed; spurious closures | double-cover shell | UDF-like shell + phase separation |
| Surface extraction | standard MC/DC on $\{s = 0\}$ | requires optimization or gradients of $u$ | standard MC on $\{\phi = 0\}$ |

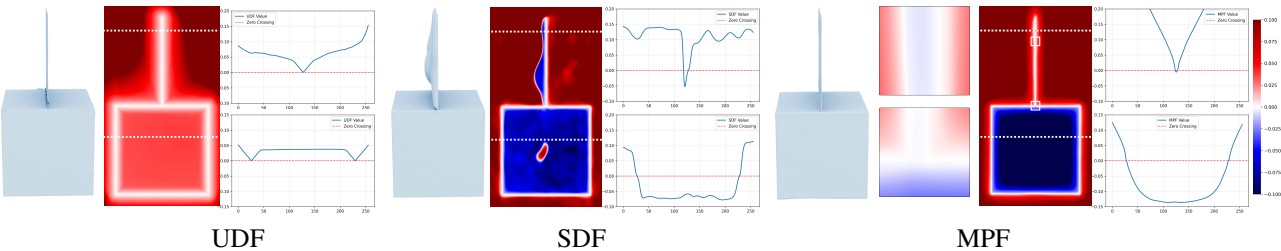

UDF           SDF           MPF

*Figure 3.* Toy example: cube + attached thin sheet (non-manifold + boundary). The shape is watertight on the cube, but includes a thin sheet attached to the top face, forming a non-manifold junction at the attachment and a boundary edge at the free end of the sheet. From left to right we show a UDF learned by CAP-UDF (Zhou et al., 2024), an SDF learned by NSH (Wang et al., 2023a), and our MPF composite field $\phi$. Each panel shows the scalar field on a 2D slice; white indicates values close to zero, while the colormap encodes sign and magnitude (warm: positive; cool: negative; note that UDF values are nonnegative). The dashed horizontal lines indicate two 1D probes (one crossing the thin sheet, one crossing the cube); the plots show the corresponding field values along these probes, and the red dashed line marks zero. UDF captures the thin sheet but cannot provide an inside–outside sign for the cube. SDF produces a clean sign change on the watertight cube but struggles near the sheet/boundary where a globally consistent sign is ill-posed. MPF preserves signed behavior on the cube while exhibiting a UDF-like narrow shell around the sheet, enabling direct extraction of $\{\phi = 0\}$ via standard Marching Cubes. Close-up views further highlight how MPF performs robustly on both watertight regions and thin-plate structures.

of UDF-based methods, including CAP-UDF (Zhou et al., 2024), GeoUDF (Ren et al., 2023), DEUDF (Xu et al., 2025) and S$^2$DF (Yang et al., 2025). These methods learn unsigned distance-like fields and are more tolerant of surfaces with boundaries, thin sheets, and other sign-ambiguous configurations, although they often require additional machinery for robust surface extraction.

**Comparison with baselines.** The ShapeNet subset contains relatively simple thin structures with smooth geometry, whereas the artist-designed and Thingi10K models are more challenging due to frequent non-manifold configurations, sharp features, and pronounced multi-scale variation, such as fine details coexisting with large smooth regions. SDF-based baselines often introduce incorrect closures on thin plates or erroneously merge nearby layers. In contrast, our method consistently produces stable reconstructions and correctly handles non-manifold configurations, thin plates, thin shells, and boundary structures. These advantages are reflected in both the quantitative and qualitative results; see Table 2 and Figures 7, 8, and 10).

UDF-based methods are better suited for modeling non-manifold configurations, boundary structures, and thin plates. However, their gradients can vanish near the zero level set or become unstable, as reported in DEUDF (Xu

et al., 2025). Since gradient information plays an important role during learning and is commonly used to guide iso-surface extraction in the absence of sign information (Zhou et al., 2024; Ren et al., 2023), such instability often leads to over-smoothed geometry or unreliable surface extraction; see Figure 9. In contrast, MPFs maintain stable gradients at the zero level set and support direct iso-surfacing with standard Marching Cubes. Consequently, our method achieves higher geometric fidelity than UDF-based baselines while remaining computationally efficient, without relying on the time-consuming optimization-based surface extraction used by DEUDF (Xu et al., 2025); see Figure 11.

**Evaluation on noisy and real-world inputs.** We further evaluate robustness under four challenging settings: indoor scenes, outdoor scenes, real-scanned objects, and synthetic noise perturbations. Specifically, we use ScanNet indoor scenes (Dai et al., 2017), outdoor scenes from the 3D Scene dataset (Zhou & Koltun, 2013), real-scanned SRB objects (Williams et al., 2019), and point clouds corrupted with synthetic Gaussian noise. For ScanNet, we evaluate both noisy point clouds reconstructed from RGB-D videos and clean points sampled from ground-truth meshes. As shown in Figure 16, SDF-based methods tend to produce bulging artifacts under noisy inputs, while UDF-based methods of-

ten suffer from fragmented surfaces. In comparison, our method consistently produces high-quality reconstructions in both settings. Figure 15 further shows scene-level results, demonstrating robustness in complex indoor environments. On the SRB dataset, which contains real-scanned noise and missing regions, our method reconstructs accurate surfaces despite incomplete observations, as shown in Figure 17. We also test synthetic Gaussian noise perturbations at 0.5%, where MPFs remain stable and preserve reasonable results.

**Controlled thin-structure analysis.** To evaluate robustness under varying geometric scales, we conduct controlled thin-structure experiments with different thickness levels $T$. Quantitative results are reported in Table 3, and the corresponding visual comparisons are shown in Figure 14. As the structures become progressively thinner, the SDF baseline, NSH, rapidly degrades and often fails to recover complete geometry, producing missing regions or severe artifacts. This behavior reflects the difficulty of maintaining a stable signed distance field near extremely thin surfaces. The UDF baseline, GeoUDF, alleviates the sign constraint but remains sensitive to local surface ambiguity, leading to topological inconsistencies such as surface splitting and holes at small thickness scales. In contrast, our method consistently preserves structural integrity across all thickness levels. Even in extremely thin regimes, MPFs maintain accurate geometry without collapse or fragmentation, demonstrating strong robustness to scale variation.

*Table 2.* Quantitative results on (top) watertight models and (bottom) open surfaces; all models contain non-manifold structures and thin sheets. We report the Chamfer Distance, scaled by $10^3$, comparing our method against SDF-based baselines. For real-world scan, the metric is computed as the average point-to-surface distance due to the absence of ground-truth geometry.

| Method | Lamp2 (100k) | Lattice (100k) | Arch (100k) | V-Box (100k) | Dolphin (10k) | Artichoke (70k) | Ship (10k) | Spikeball (140k) |
|---|---|---|---|---|---|---|---|---|
| NSH | 1.162 | 4.046 | 2.149 | 1.224 | 0.454 | 5.287 | 5.591 | 1.329 |
| DiGS | 1.491 | 3.646 | 2.021 | 1.190 | 0.878 | 7.963 | 7.945 | 6.074 |
| StEik | 1.198 | 2.250 | 0.591 | 1.047 | 0.515 | 5.442 | 2.596 | 7.506 |
| PG-SDF | 1.766 | 6.631 | 0.940 | 1.101 | 0.436 | 1.335 | 2.426 | 1.480 |
| I-filtering | 1.160 | 21.354 | 2.912 | 1.551 | 0.776 | 3.881 | 4.977 | 2.120 |
| Ours | 1.106 | 2.012 | 0.506 | 0.696 | 0.401 | 1.127 | 0.672 | 0.982 |

| Method | Flower (70k) | Lamp1 (70k) | Dress (10k) | Cloth (10k) | Vase (10k) | Hat (10k) | Insertion (10k) | Real-scan (3M) |
|---|---|---|---|---|---|---|---|---|
| NSH | 10.425 | 19.387 | 9.502 | 37.164 | 7.732 | 7.047 | 9.459 | 1.435 |
| DiGS | 6.237 | 8.851 | 7.344 | 4.663 | 9.535 | 10.09 | 12.834 | 0.726 |
| StEik | 7.729 | 24.334 | 7.777 | 5.470 | 13.263 | 8.028 | 20.823 | 0.695 |
| PG-SDF | 9.036 | 3.530 | 15.481 | 10.915 | 17.25 | 26.58 | 18.198 | 10.118 |
| I-filtering | 45.648 | 2.457 | 4.976 | 3.698 | 2.975 | failed | 17.721 | 19.339 |
| Ours | 1.757 | 1.165 | 2.113 | 2.194 | 2.625 | 1.534 | 1.807 | 0.678 |

**Ablation studies.** We isolate the contribution of each core loss component by either removing the corresponding term or increasing its weight by $10\times$. The results are summarized in Table 4, with qualitative comparisons shown in Figure 13. Removing key loss terms substantially degrades reconstruction quality, especially for thin structures and open-boundary shapes. In contrast, increasing individual

*Table 3.* Quantitative comparison of SDF-based (NSH), UDF-based (GeoUDF), and our MPF method on thin structures under different thickness scales $T$. SDF-based methods fail or produce severe artifacts at extremely thin scales, UDF-based methods suffer from topological issues such as splitting or holes, while MPF consistently achieves accurate reconstruction.

| Thickness ($T$) | SDF (NSH) | UDF (GeoUDF) | Ours (MPF) |
|---|---|---|---|
| 0.10 | 0.40 (artifact) | 0.42 (splitting) | 0.26 |
| 0.05 | 0.26 | 0.43 (splitting) | 0.27 |
| 0.01 | 3.22 (failure) | 0.42 (holes) | 0.20 |
| 0.005 | 4.04 (failure) | 0.35 (holes) | 0.23 |

*Table 4.* Ablation study on MPF loss components. We either remove a term or increase its weight by $10\times$. Each component contributes to stable optimization and improved reconstruction quality, especially for thin structures.

| $\lambda_{\text{align}}$ | $\lambda_{\text{normal}}$ | $\lambda_{\text{lap}}$ | $\lambda_{\text{phase}}$ | Description | CD $\downarrow$ |
|---|---|---|---|---|---|
| $1\times$ | $1\times$ | $1\times$ | $1\times$ | Default | 1.99 |
| $10\times$ | $1\times$ | $1\times$ | $1\times$ | Strong alignment | 2.31 |
| $0$ | $1\times$ | $1\times$ | $1\times$ | No alignment | 6.29 |
| $1\times$ | $10\times$ | $1\times$ | $1\times$ | Strong normal constraint | 2.57 |
| $1\times$ | $0$ | $1\times$ | $1\times$ | No normal constraint | 2.33 |
| $1\times$ | $1\times$ | $10\times$ | $1\times$ | Strong Laplacian constraint | 2.17 |
| $1\times$ | $1\times$ | $0$ | $1\times$ | No Laplacian constraint | 4.19 |
| $1\times$ | $1\times$ | $1\times$ | $10\times$ | Strong phase constraint | 2.08 |
| $1\times$ | $1\times$ | $1\times$ | $0$ | No phase constraint | 10.68 |

weights leads to only minor performance variations, indicating that the proposed objective is robust to moderate hyperparameter changes and that the loss terms are well balanced.

# 6. Conclusion

We introduced Metric–Phase Fields, a new type of implicit function for reconstructing surfaces with thin structures, surfaces with boundaries, and non-manifold configurations from unoriented point clouds. MPFs separate metric proximity and sign/phase information by learning an unsigned metric field $r$ alongside a smooth phase field $\theta$, and combining them through a gated-metric formulation with residual phase injection to create a signed composite field that supports stable optimization and direct iso-surface extraction. This design maintains SDF-like signed behavior where an inside–outside notion is meaningful while smoothly showing UDF-like behavior in sign-ambiguous regions such as thin plates and boundary edges. MPFs enable direct zero level set extraction via the standard Marching Cubes algorithm. Across synthetic and scanned benchmarks, MPFs more faithfully preserve thin and layered geometry than recent SDF-based baselines, and provide more reliable training and extraction than UDF-based alternatives.

## Acknowledgment

This research is supported by cash and in-kind funding from NTU S-Lab and industry partner(s). It is also partially supported by the Ministry of Education, Singapore, under its Academic Research Fund Grant RT19/22, the Research Projects of ISCAS (ISCAS-JCMS-202303, ISCAS-ZD-202401, ISCAS-JCZD-202402 and ISCAS-JCMS-202403), and in part by the Hong Kong Research Grants Council under Grants 11219324 and N_CityU1114/25.

## Impact Statement

This paper presents work whose goal is to advance the field of Machine Learning. There are many potential societal consequences of our work, none which we feel must be specifically highlighted here.

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

## A. Extended Analysis

**Effect of Scaling Factor $\beta$.**   The parameter $\beta$ controls the sharpness of the phase transition by scaling the magnitude of the phase field $\theta$ near the surface, thereby influencing both the strength of gradient signals and the convergence behavior during training. We evaluate its effect on thin-walled structures using $\beta \in 1, 50, 100, 500$. For each setting, we visualize the reconstructed meshes at different training stages (1k, 4k, and 10k iterations).

As shown in Fig. 4, a small value of $\beta$ ($\beta = 1$) leads to slow convergence, and the model has not fully converged even after 10k iterations. While the model can already recover a reasonable surface at early iterations due to the large learning amplitude of $\theta$, subsequent refinement progresses gradually. A moderate value of $\beta = 50$ achieves the best trade-off, enabling fast and stable convergence and producing clean, topologically consistent thin-plate geometries. When $\beta$ is increased to 100, the optimization becomes less stable at early iterations but converges more rapidly in later stages. In contrast, an excessively large value ($\beta = 500$) results in unstable training and fails to produce meaningful reconstructions.

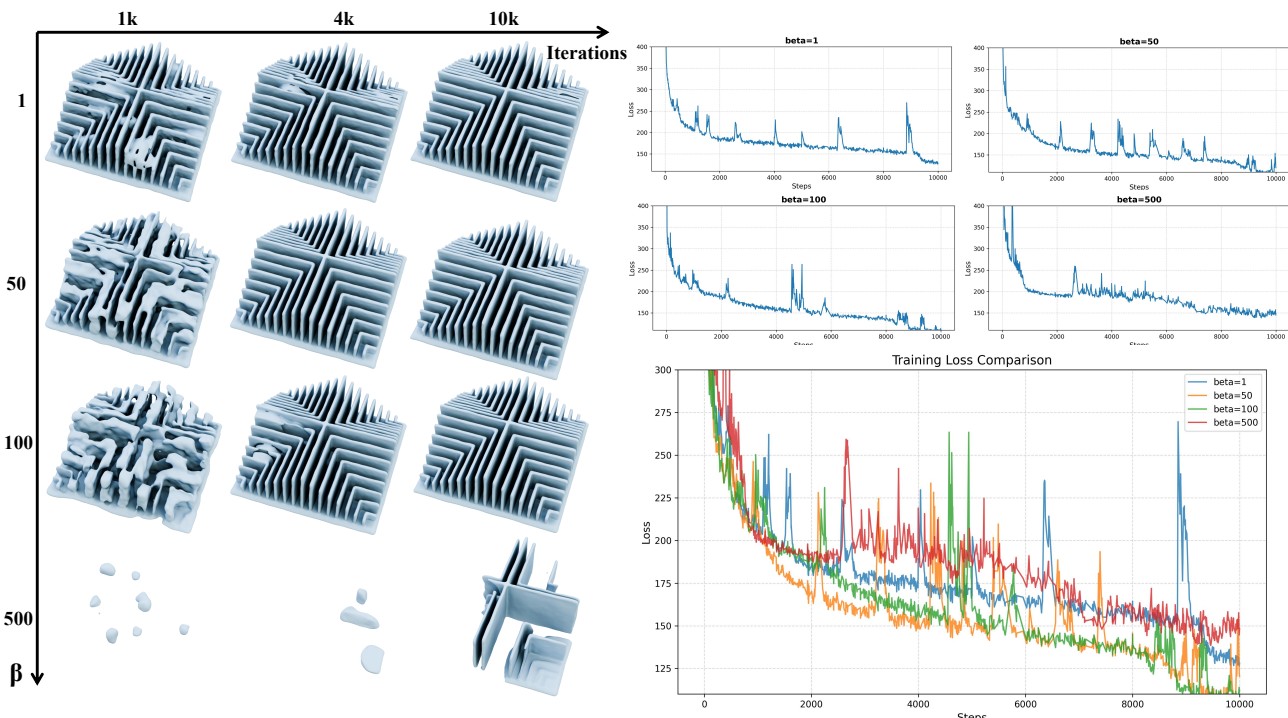

*Figure 4.* Ablation study of the phase scaling parameter $\beta$ on thin-walled structures. Results are shown for $\beta \in \{1, 50, 100, 500\}$ at 1k, 4k, and 10k training iterations.

**Effect of directly parameterizing $\phi(x) = P(x)$.**   We investigate a simplified formulation that directly learns the implicit field as $\phi = \tanh(\beta\,\theta(\mathbf{x}))$, without explicitly modeling a distance component. Although this formulation can partially recover thin-shell structures under carefully chosen $\beta$, we observe several fundamental limitations.

Due to the saturation property of the hyperbolic tangent, the gradient of $\phi$ is highly localized near the zero level set of $\theta$. Specifically, $\nabla\phi = \beta\big(1 - \tanh^2(\beta\theta)\big)\nabla\theta$ vanishes rapidly away from the surface. As a result, gradient-based constraints such as the Eikonal loss only affect an infinitesimal neighborhood around the surface, preventing sign information from propagating into the surrounding space. This often leads to degenerate solutions where the implicit field collapses to a single sign away from the surface, effectively enclosing the shape.

We further find that the choice of $\beta$ is highly sensitive. A small $\beta$ enlarges the effective gradient region and can alleviate sign collapse for thin-shell structures, but tends to introduce high-frequency oscillations and unstable optimization. In contrast, a large $\beta$ overly localizes the gradient near the surface, making sign flipping difficult to learn and resulting in unstable or trivial solutions, as shown in Figure 5.

More critically, for thin-plate geometries with open boundaries, this parameterization fundamentally fails to represent valid

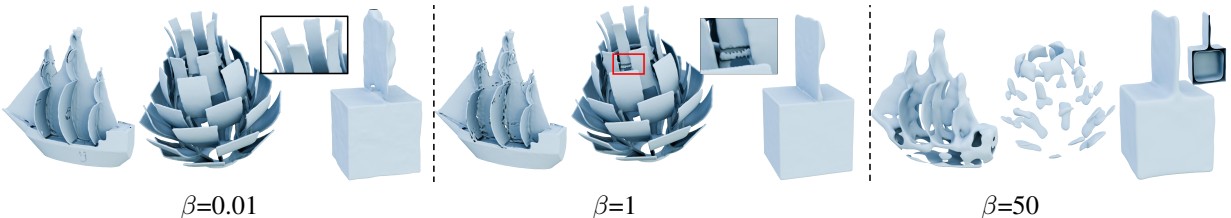

$\beta$=0.01          $\beta$=1          $\beta$=50

*Figure 5.* Result obtained when the shape is represented solely by the phase term, i.e., $\phi(x) = P(x)$.

implicit surfaces. Since $\phi = \tanh(\beta\theta)$ enforces continuous sign transitions everywhere, it cannot correctly terminate a single surface at boundary regions, inevitably producing spurious thin layers near the boundaries. These observations highlight the necessity of explicitly decoupling distance and sign modeling, which enables stable gradient propagation and flexible representation of open and non-manifold geometries.

## B. Discussions

We discuss key properties of MPFs and explain why they are useful for reconstructing surfaces with thin structures and boundaries.

**Why decouple metric and topology?** Learning an SDF directly from unoriented points requires the network to recover the boundary surface (metric) and provide a consistent inside–outside labeling (topology) at the same time. These objectives interact: an incorrect topological guess early in training can distort geometric convergence and lead to poor local minima. MPFs separate the learning into two complementary tasks: (i) Metric stability: $r(\mathbf{x})$ captures proximity without being affected by sign ambiguity; and (ii) Phase learning: $\theta(\mathbf{x})$ encodes a soft inside–outside preference when such a notion is meaningful. The gated-residual coupling in (3) lets the two tasks support each other without collapsing into a single ill-conditioned objective.

**If $\theta$ defines the interface, why does $r$ still matter?** Under the mild structural condition $r(\mathbf{x}) \geq 0$ (which can be enforced by design, e.g., a nonnegative activation), Proposition 3.1 implies $\{\phi = 0\} = \{\theta = 0\}$. In this regime, $r$ does not "move" the interface; instead it provides stable metric supervision off the surface (via $\mathcal{L}_{\text{eik}}^r$) and shapes the behavior of $\phi$ away from the interface through the gated term $rP$. This improves conditioning compared to using a purely phase-driven signed field.

**Why does the residual stabilize training?** In a purely multiplicative construction $\phi = rP$, the backpropagated gradient to the phase branch is scaled by $r(\mathbf{x})$:

$$\frac{\partial \phi}{\partial \theta} = r(\mathbf{x})\,\beta\,\text{sech}^2\big(\beta\,\theta(\mathbf{x})\big), \tag{19}$$

which collapses near the interface where $r(\mathbf{x}) \approx 0$. With the residual term, we obtain

$$\frac{\partial \phi}{\partial \theta} = 1 + r(\mathbf{x})\,\beta\,\text{sech}^2\big(\beta\,\theta(\mathbf{x})\big) \approx 1 \quad \text{when } r(\mathbf{x}) \approx 0, \tag{20}$$

providing a direct and non-vanishing gradient pathway for learning the phase transition.

**Why MPF improves thin-structure reconstruction.** For standard watertight geometries (e.g., Bunny, Armadillo), SDF-based methods already achieve strong performance, and MPF yields comparable results. Our primary focus is on more challenging regimes involving thin structures and tightly coupled surface layers, where SDF-based methods often degrade, as shown in Table 5.

This performance gap arises from the coupled nature of sign and distance in standard SDF learning. The network must simultaneously model surface proximity and consistent inside–outside labeling. In thin or tightly enclosed regions, this coupling makes it difficult to represent rapid sign transitions within a narrow spatial band, often leading to thickness inflation, surface merging, or spurious closures.

In contrast, MPF decouples metric and phase representations. The metric field captures unsigned proximity under standard geometric regularization, while the phase field independently models sign transitions. Since the phase component is

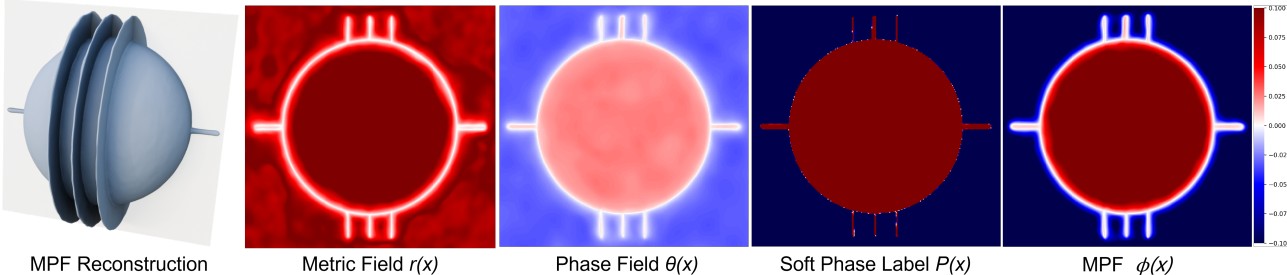

| MPF Reconstruction | Metric Field $r(x)$ | Phase Field $\theta(x)$ | Soft Phase Label $P(x)$ | MPF  $\phi(x)$ |

*Figure 6.* Visualization of the components of our MPF representation. Separately showing these fields highlights their complementary contributions to the final reconstruction.

not constrained by unit-gradient conditions, it can better adapt to rapid sign changes without interfering with distance learning. This results in a better-conditioned optimization process and improved reconstruction quality for thin and complex geometries.

*Table 5.* Quantitative comparison using Chamfer Distance (CD ↓). On standard watertight models such as Bunny and Armadillo, all methods perform comparably. On challenging thin-structure cases (e.g., Ship), MPF significantly outperforms SDF- and UDF-based baselines, demonstrating its effectiveness in handling thin and complex geometries.

| Model | DiGS | StEik | PG-SDF | I-filtering | NSH | Ours |
|---|---|---|---|---|---|---|
| Bunny | 0.454 | 0.652 | 0.396 | 0.503 | 0.494 | 0.458 |
| Armadillo | 0.506 | 0.558 | 0.481 | 0.550 | 0.422 | 0.476 |
| Ship (thin structures) | 7.945 | 2.596 | 2.426 | 4.997 | 5.591 | 0.672 |

**Behavior on surfaces with boundary and non-manifold regions.** In regions where a globally consistent inside–outside labeling is ambiguous or ill-posed (e.g., thin fins, surfaces with boundary, and certain non-manifold configurations), the metric branch $r$ remains well-defined and behaves like a UDF, encouraging a stable geometric proxy. Empirically, this often manifests as a UDF-like shell that wraps boundary edges (a "double cover" effect), while the phase field $\theta$ separates the two sides by taking distinct values across the wrapped proxy surface.

In downstream applications, watertight and stable surface representations are often preferred for compatibility with standard processing pipelines, while explicitly open surfaces may lead to fragmented geometry.

Overall, MPF represents such regions as implicit thin shells, providing a practical trade-off between geometric flexibility and downstream usability without requiring explicit boundary handling.

**Surface extraction.** For Marching Cubes extraction, we use a fixed high-resolution grid for all experiments, which is sufficient for capturing thin structures in most cases (e.g., plant in Fig. 1). For scenarios involving highly anisotropic or extremely thin geometries (e.g., thin plates in Fig. 14), the grid resolution is adjusted according to the smallest spatial extent of the object to ensure adequate sampling density. Alternatively, adaptive octree-based Marching Cubes methods can further improve efficiency by refining cells only in regions where the implicit field exhibits high variation, making them a lightweight option for handling extremely fine structures.

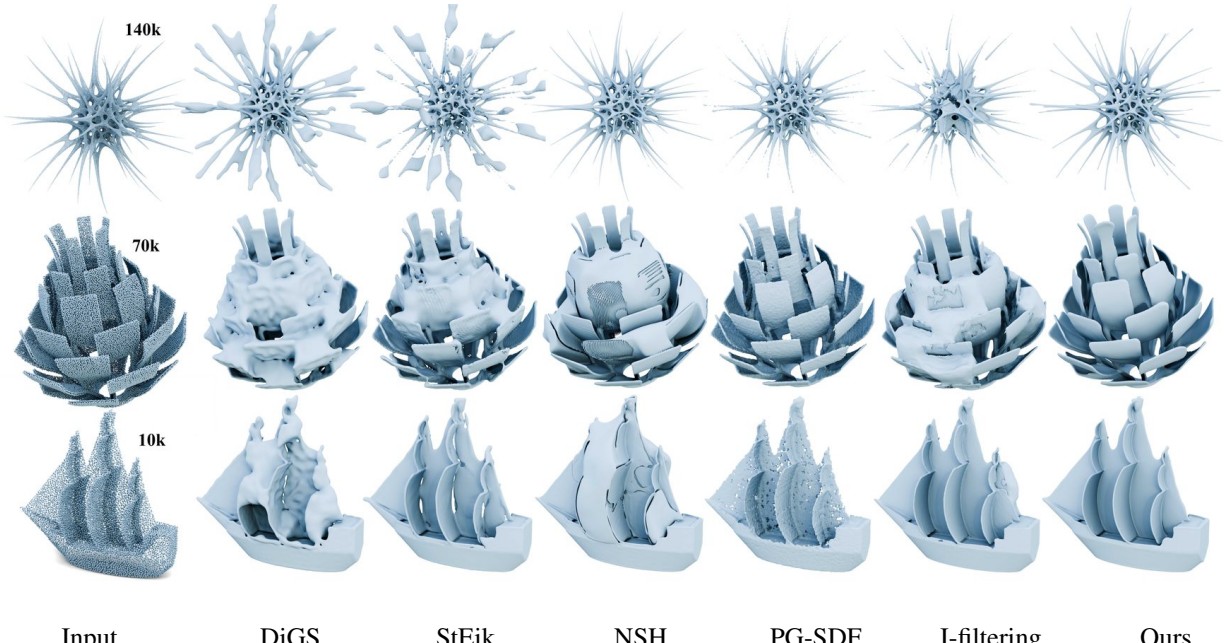

| Input | DiGS | StEik | NSH | PG-SDF | I-filtering | Ours |

*Figure 7.* Qualitative comparison on watertight models (Spikeball, Artichoke, Ship) with complex thin and multi-layered structures under varying sampling levels. Compared to DiGS (Ben-Shabat et al., 2022), StEik (Yang et al., 2023), NSH (Wang et al., 2023a), PG-SDF (Koneputugodage et al., 2024), and I-filtering (Li et al., 2025), our approach produces significantly more robust reconstructions, especially in the presence of intricate thin structures and layered geometries.

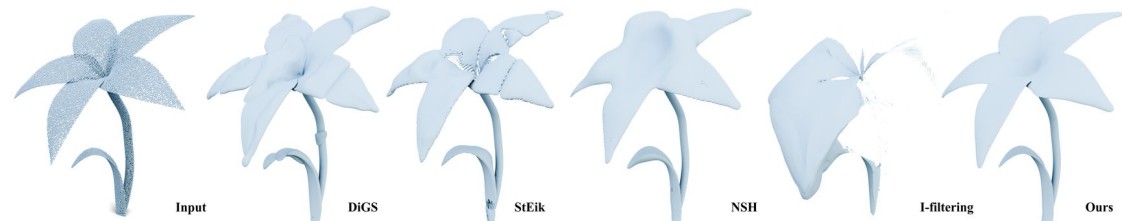

| Input | DiGS | StEik | NSH | I-filtering | Ours |

*Figure 8.* Qualitative comparison on inputs that consist of *single-layer* point clouds. Compared to DiGS (Ben-Shabat et al., 2022), StEik (Yang et al., 2023), NSH (Wang et al., 2023a), and I-filtering (Li et al., 2025), our method produces thinner reconstructions that more closely adhere to the original input point clouds.

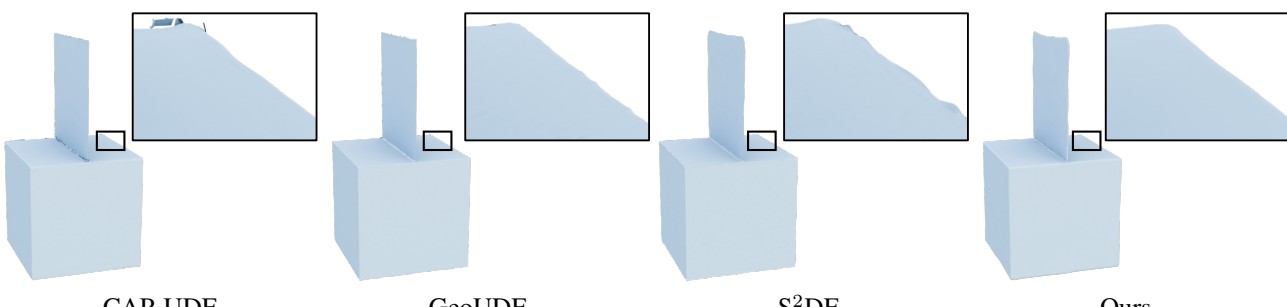

| CAP-UDF | GeoUDF | S²DF | Ours |

*Figure 9.* Qualitative comparison on a toy model containing boundary and non-manifold structures. We use this toy model with simple geometry to evaluate iso-surface extraction behavior. All UDF-based baselines are able to recover the overall shape reasonably well. However, they either rely on gradient information or optimization-based procedures to extract the zero level set. Small inaccuracies in UDF values can lead to large errors in the estimated gradients (e.g., gradient direction flips), which in turn cause artifacts in the extracted surfaces. As a result, both CAP-UDF (Zhou et al., 2024) and GeoUDF (Ren et al., 2023) exhibit noticeable non-smoothness in their iso-surfaces. $S^2DF$ (Yang et al., 2025), in contrast, employs DCUDF (Hou et al., 2023), an optimization-based surface extraction method, which produces better results than CAP-UDF and GeoUDF. Nevertheless, minor non-smooth artifacts remain. Moreover, its iso-surfacing stage is significantly more time-consuming than ours: at a resolution of $256^3$, $S^2DF$ requires 96 seconds, whereas our method takes only 5 seconds.

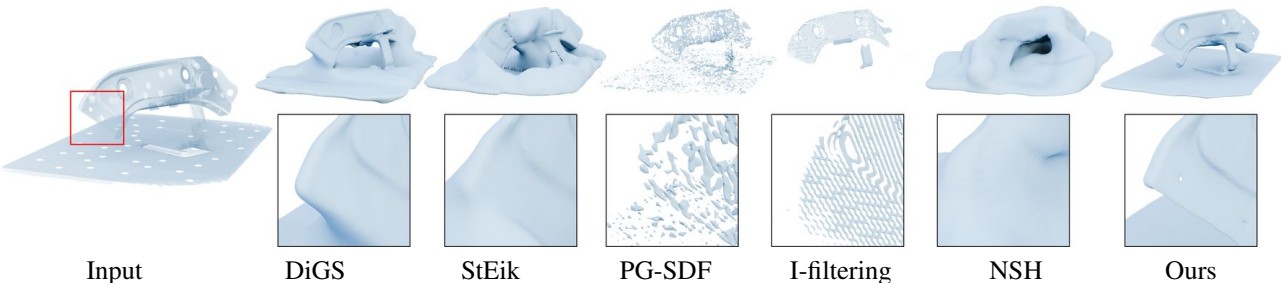

| Input | DiGS | StEik | PG-SDF | I-filtering | NSH | Ours |

*Figure 10.* Comparisons on a real-scan thin-structure model. The real-scan model contains approximately 3 million input points with thin structures, which are challenging to reconstruction from real scans. Results are extracted at a resolution of $512^3$.

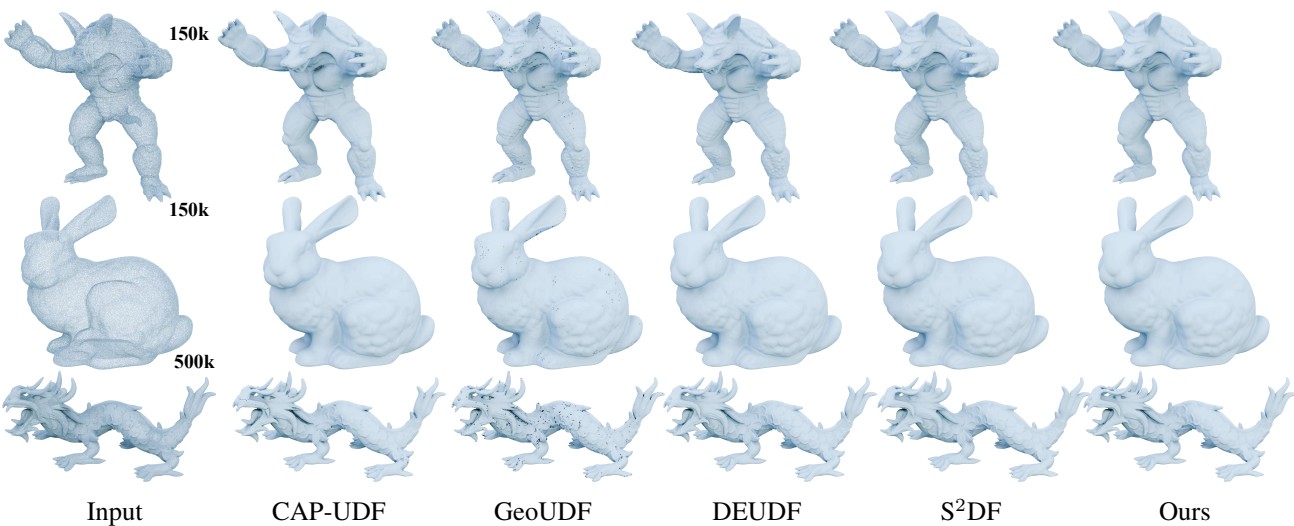

| Input | CAP-UDF | GeoUDF | DEUDF | $S^2DF$ | Ours |

*Figure 11.* Comparison with UDF-based baselines on graphics benchmarks featuring watertight geometries with fine details. While all methods produce visually plausible reconstructions, our method demonstrates clear advantages in zero level set extraction. For example, on the Dragon model at a resolution of $512^3$, extracting the zero level set takes 10.5, 3.6, 5.8, and 15.2 minutes for CAP-UDF, GeoUDF, DEUDF, and $S^2DF$, respectively. In contrast, our method relies on standard Marching Cubes and requires only 0.13 minutes, significantly outperforming the UDF-based baselines.

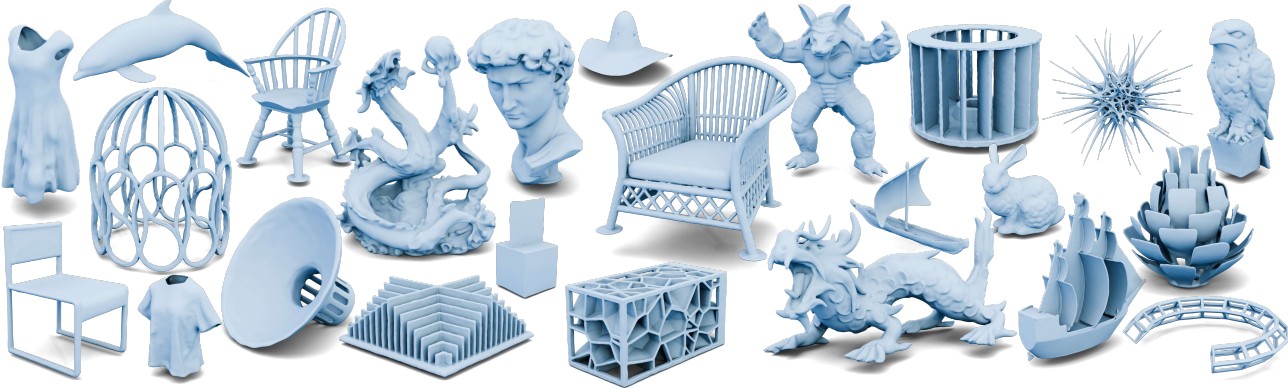

*Figure 12.* MPF gallery: reconstructions across diverse geometries and topologies, including thin structures and open boundaries that challenge SDF-based methods.

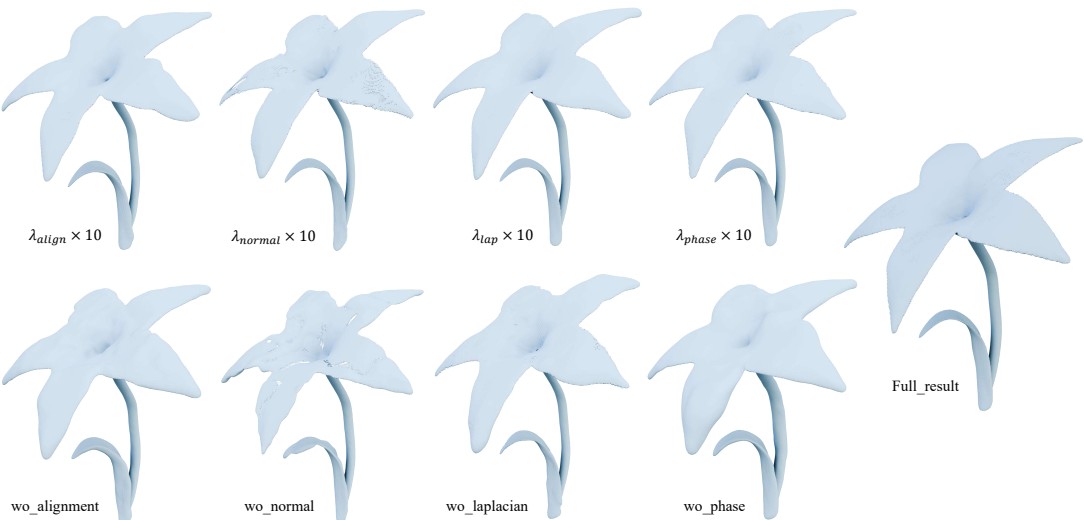

*Figure 13.* Visual comparison for the ablation study in Table 4. Removing the normal constraint leads to unstable surface orientations and degraded reconstruction quality, despite only minor changes in CD.

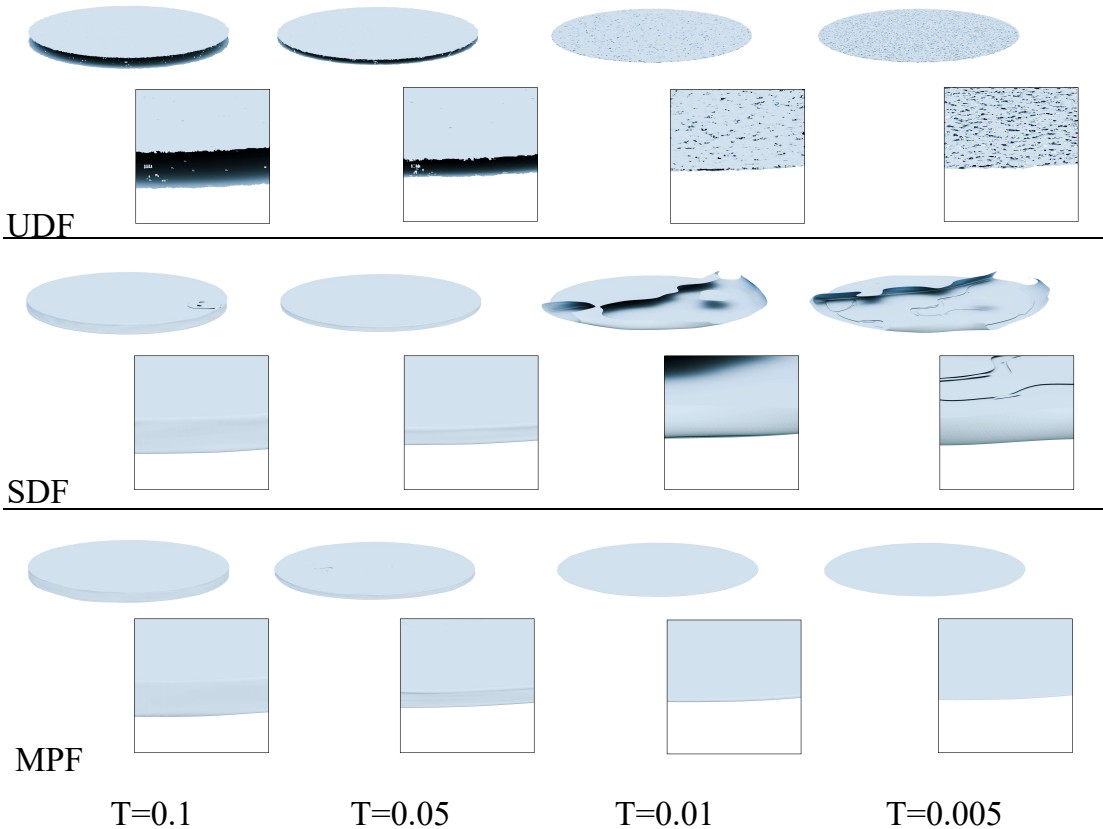

*Figure 14.* Qualitative comparison of thin-structure reconstruction at varying thickness levels $T$. SDF-based methods fail to preserve extremely thin geometry, while UDF-based methods suffer from topological artifacts such as holes and splitting. In contrast, MPF consistently reconstructs accurate and complete thin structures across all scales.

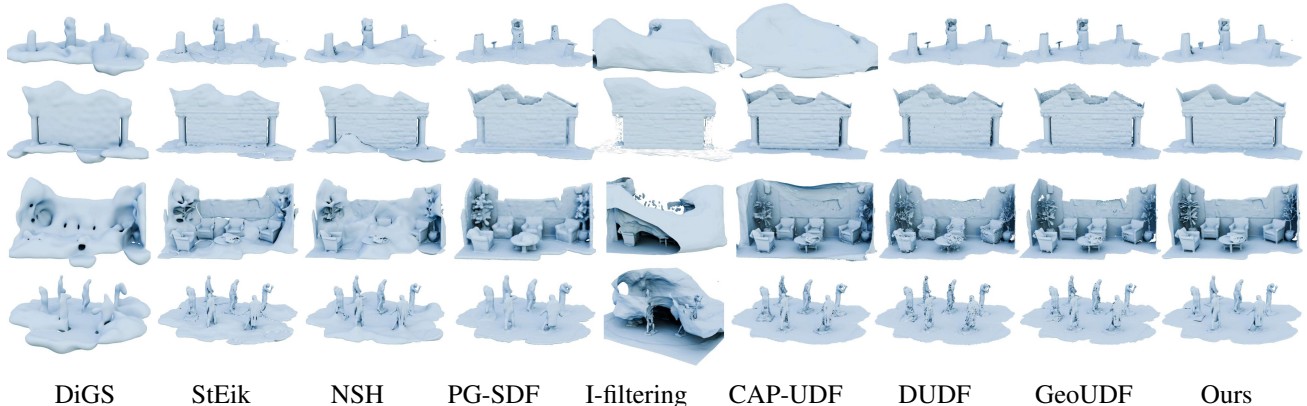

DiGS  StEik  NSH  PG-SDF I-filtering CAP-UDF DUDF GeoUDF Ours

*Figure 15.* Comparison of different methods on the scene datasets. SDF-based methods include DiGS, StEik, NSH, I-filtering and PG-SDF; UDF-based methods include CAP-UDF, DUDF, and GeoUDF. Our method demonstrates superior performance in terms of reconstruction quality.

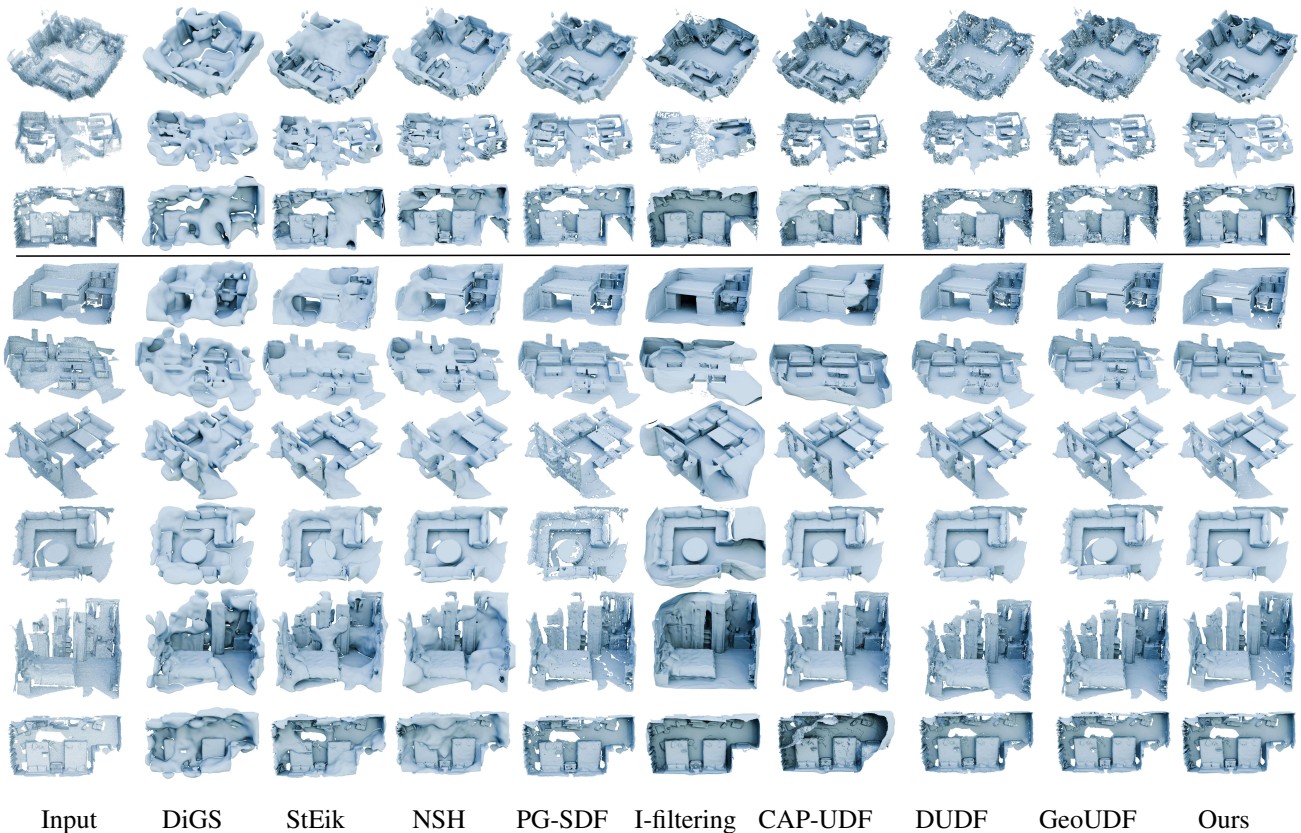

Input  DiGS  StEik  NSH  PG-SDF I-filtering CAP-UDF DUDF GeoUDF Ours

*Figure 16.* Comparison of indoor scene reconstruction results. The first three rows show point maps extracted from RGB-D videos in ScanNet, which contain significant noise, while the subsequent rows use points sampled from the official meshes as input. SDF-based methods tend to produce bulging artifacts, I-filtering achieves good results, and UDF-based methods are prone to fragmented surfaces. Our method produces high-quality reconstructions in both cases. Note that I-filtering and CAP-UDF produce a watertight envelope of the scene. For better interior visualization, a cropping post-processing step was performed on the reconstructed manifold.

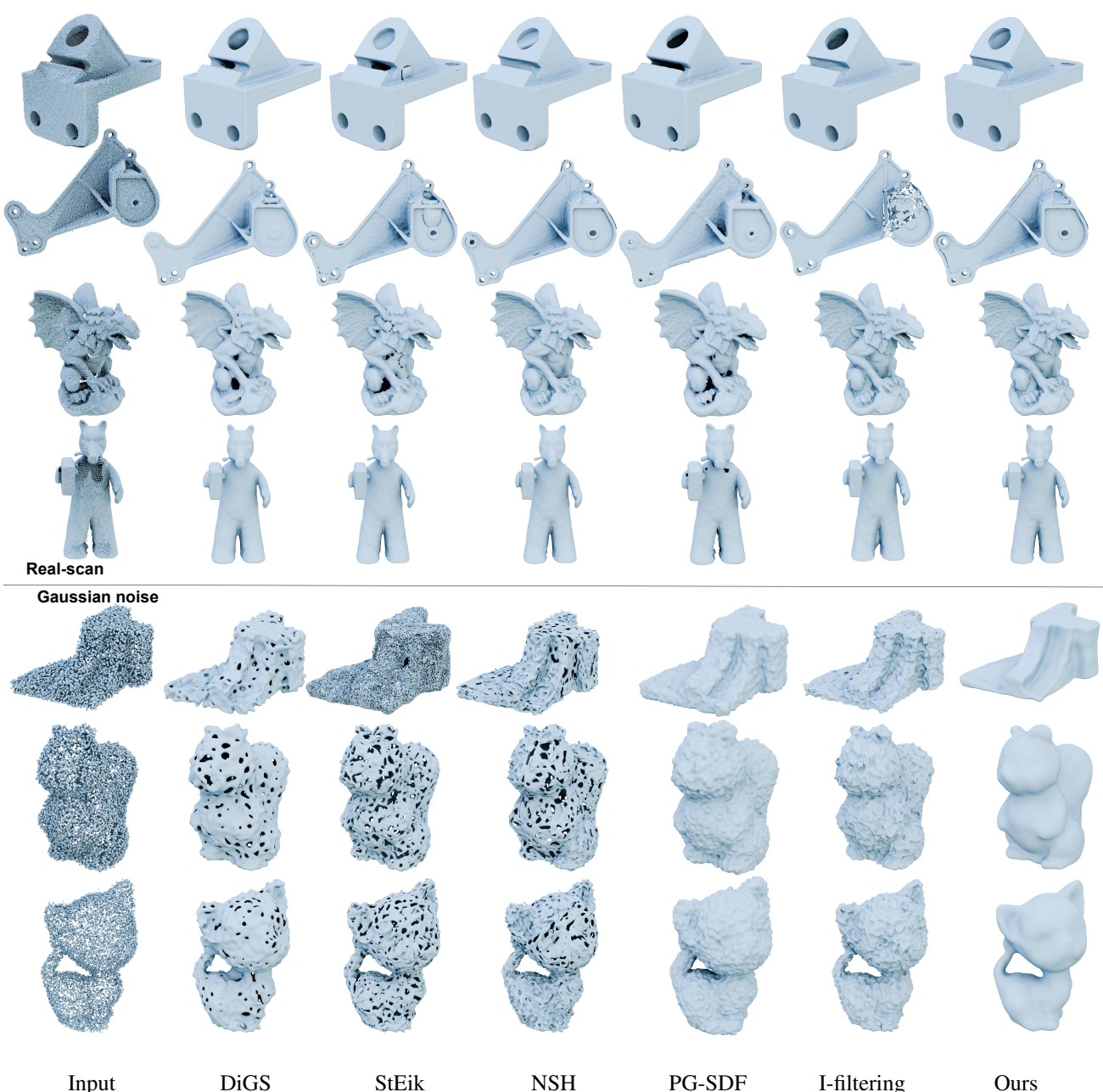

| Input | DiGS | StEik | NSH | PG-SDF | I-filtering | Ours |

*Figure 17.* The top four rows compare different methods on the SRB dataset, which contains partial missing regions and noise/outlier perturbations due to its real-scanned nature. Our method achieves accurate reconstructions despite these challenges. The bottom three rows show reconstructions with added Gaussian noise (0.5%), highlighting the robustness of our method under noisy conditions. All methods are evaluated using their default settings.

