# OpenReview forum: "Metric—Phase Fields: Decoupling Distance and Sign for Thin-Structure Reconstruction from Unoriented Point Clouds"
_ICML.cc/2026/Conference — ICML 2026 regular_

### Official Review · Reviewer_N1Uv · 2026-03-02

**Soundness:** 3
**Presentation:** 3
**Significance:** 3
**Originality:** 2
**Overall Recommendation:** 4
**Confidence:** 4

**Summary:**

This submission introduces a way of computing implicit fields for surface reconstruction by decoupling the sign and distance. More precisely, it relies on an inner/outer smooth classification and an unsigned distance field. The final field is reconstructed by combining the two and injecting a residual gradient ensuring that the gradient does not vanish near the surface.
Usual losses are used (Dirichlet near the surface, Neumann if normals  -- possibly unoriented -- are available), as well as some specifics (eikonal near the surface for the unsigned distance field $r$ , and laplacian regularization away from the surface on $\phi$, saturation promoting the smooth classification to be +1/-1 outside of a narrow band, and a far field repulsion with an exponential barrier). This is implemented with a SIREN backbone network and then a geometry head separated from a sign head.

**Compliance With Llm Reviewing Policy:**

Affirmed.

**Final Justification:**

I raised my score to weak accept following the rebuttal (see rebuttal acknowledgment for the reasons).

**Key Questions For Authors:**

Q1 - The extraction with marching cubes in case of very thin objects would require sampling and extremely fine grid. Is this the case? How can we be sure to recover the surface sheet (except by trial and error)?

Q2 - Regarding this marching cube resolution, wouldn't it create hole in the reconstruction? can we guarantee a width of the thin shell that would allow to guess the marching cube resolution?

Q3 - How does the method compare if using a 1-lipschitz network for the unsigned distance backbone.

**Limitations:**

No societal impact issue.

**Strengths And Weaknesses:**

Strengths

- I found the idea of decoupling the sign part from the unsigned distance field interesting, also it is not entirely new (Erler 2020)

- The residual gradient injection makes a lot of sense to stabilize the training, and is well founded

Weaknesses

- Point2surf (Points2Surf: Learning Implicit Surfaces from Point Clouds, Erler 2020) also has a sign/UDF separation in their scheme, the authors should at least comment upon this method.

- The behavior for open surfaces is not very clear (or thin plate surface: surface made of a single sheet without the possibility to define an exterior or an interior. From the text and the figures, it appears that this results in a thin shell surface, a *closed* surface enclosed within two very close sheets. This means that the method does not preserve the characteristic of the object.

- The extraction with marching cubes in case of very thin objects is not very clear (or, if I understand correctly, thin shell transformed from thin-plate parts) would require sampling and extremely fine grid. Is this the case? How can we be sure to recover the feature (except by trial and error)?

- There is an issue in section 3.3: Proposition 3.2 explains that for training sample the constraints are enforced which is clearly incorrect since the constraint is only promoted through a loss (surface data loss), and in many cases the loss will not converge to 0.

- I would also recommend exploring and comparing against 1-lipschitz network which can be used for UDF or SDF and also allow to recover thin structures (by converting from thin-plate to thin-shell). By promoting the lipschitz constant to be closed to 1, they also remove any gradient instability near the surface. [Coiffier, Bethune 2023]

remarks:
line 200 explicidtly
line 227 small constant $\varepsilon$ not used
- Notation of the norm consistency between equations (1) and (10)

---

> ### Author Rebuttal · Authors · 2026-03-31
>
> We thank the reviewer for the insightful comments. We have included additional quantitative comparisons and visualizations in the [Anonymous Rebuttal Link](https://anonymous.4open.science/r/ICML-rebuttal-20018/rebuttal.md), and we would like to clarify several points below.
>
> 1. Marching Cubes extraction for thin structures and potential holes (Q1 & Q2).
>
> We appreciate the reviewer’s concern. In our experiments, we use a fixed high resolution (e.g., $512^3$) for extracting thin structures, which suffices for most cases (e.g., the plant in Fig. 1). For highly anisotropic or extremely thin regions  (e.g., thin plate in link Figure 9), the grid is set according to the smallest spatial axis.
> In practice, octree-based Marching Cubes methods (e.g., [1],[2]) provide a lightweight alternative: leaf nodes are adaptively refined by checking the SDF values at the eight corners, increasing resolution only where needed. This approach allows reliable capture of fine structures in the vast majority of cases, and could be easily incorporated if extraction issues arise.
>
> 2. Comparison with 1-Lipschitz network for the unsigned distance backbone.
>
> Following this advice, we conducted additional experiments by incorporating a 1-Lipschitz (Coiffier and Béthune, 2023) into our unsigned distance backbone.
> We have included quantitative comparisons (link Figure 10), where we evaluate several models using both the original MPF architecture and its 1-Lipschitz variant, reporting Chamfer Distance. Our observations are as follows:
> The 1-Lipschitz variant provides improved gradient stability and performs robustly on complex topologies and open-boundary regions, showing comparable results to MPF in these scenarios.
> For closely spaced thin layers, the added smoothness from the 1-Lipschitz constraint may slightly affect the fine geometric details, such as minor surface adhesion. For surfaces with high-frequency details (e.g., the David head model), it may result in somewhat less precise reconstruction of fine features.
> Overall, Chamfer Distance does not show a clear advantage in the tested cases, suggesting that our original MPF architecture already captures thin structures effectively.
> Future work could explore Lipschitz constraints combined with adaptive tuning to further improve robustness and fidelity.
>
> 3. Points2Surf sign/UDF separation.
>
> We thank the reviewer for this suggestion. We will include a citation to Points2Surf in the revised manuscript and clarify the differences. Unlike Points2Surf, which decodes both sign and distance from a shared feature space, we learn them separately and introduce a soft constraint to encourage their alignment along the gradient direction. Concretely, this constraint promotes consistency between the distance field gradient and the surface normal implied by the sign. This improves numerical stability near the surface, where small inconsistencies can otherwise lead to artifacts. In practice, it helps eliminate the “bumpy surface” issue reported in Points2Surf and yields smoother reconstructions.
>
> 4. Behavior for open surfaces (thin-plates).
>
> From an application perspective, most downstream tasks—such as 3D printing, fluid simulation, collision detection in games, or VR/AR—require closed, manifold surfaces. Methods that natively support open boundaries often produce fragmented or unstable geometry, which typically requires extensive manual repair before it can be used in a production pipeline.
>
> By representing open or under-constrained regions as closed thin shells, MPF provides a practical and robust solution: the reconstructed surfaces remain topologically closed and usable, while still accurately capturing thin structures. This design choice balances geometric fidelity with downstream usability, reducing errors and post-processing effort.
>
>
> 5. Proposition 3.2 & minor remarks.
>
> We thank the reviewer for these comments. We will clarify the role of the surface data loss and that convergence to zero is not guaranteed, and fix notation inconsistencies and unused constants in the revision.
>
> [1] Schaefer, Scott, and Joe Warren. Dual marching cubes: Primal contouring of dual grids. 12th Pacific Conference on Computer Graphics and Applications, 2004. PG 2004. Proceedings. Ieee, 2004.
>
> [2] Shu, Renben, Chen Zhou, and Mohan S. Kankanhalli. Adaptive marching cubes. The Visual Computer, 11(4), 202-217.

---

> > ### Author Rebuttal · Reviewer_N1Uv · 2026-04-02
> >
> > Thank you for these answers and additional experiments. Overall, I find that the results are good and well demonstrated in the additional experiments. I am not completely convinced that the grid resolution issue can be discarded so easily, but given the additional experiments and comparisons, I agree to raise my score to a weak accept.

---

> > > ### Author Response · Authors · 2026-04-02
> > >
> > > Dear Reviewer N1Uv,
> > >
> > > We sincerely thank you for the positive feedback and for acknowledging the additional experiments and analyses. We are glad that the new results help clarify the effectiveness of our method.
> > >
> > > We also greatly appreciate your valuable suggestions, which are very helpful in improving our paper. Based on your comments, we will revise the manuscript in the following aspects:
> > >
> > > - Clarify related work (Points2Surf):
> > > We will include a citation to Points2Surf in the revised manuscript and clearly articulate the differences from our approach.
> > > - Improve explanation of thin-structure extraction:
> > > We will further clarify our Marching Cubes extraction strategy for thin structures, including the use of sufficiently high resolution, and discuss adaptive alternatives such as octree-based methods to support reliable recovery of fine details.
> > > - Clarify design choice for open surfaces:
> > > We will better explain the practical usefulness of reconstructing open surfaces as thin shells.
> > > - Proposition 3.2 & minor remarks, and future work:
> > > We will clarify that convergence of the surface data loss to zero is not guaranteed, and also fix notation inconsistencies and other minor issues. We will also note that future work could explore Lipschitz constraints with adaptive tuning to further improve robustness and fidelity.
> > >
> > > Thank you again for your time and the constructive feedback you have provided to improve our paper!
> > >
> > > Best regards,
> > >
> > > The Authors

---

### Official Review · Reviewer_ZxGX · 2026-03-05

**Soundness:** 3
**Presentation:** 3
**Significance:** 3
**Originality:** 3
**Overall Recommendation:** 4
**Confidence:** 5

**Summary:**

This paper focuses on improving surface reconstruction quality for thin structures. It proposes Metric–Phase Fields (MPFs) to combine the complementary strengths of UDFs and SDFs. Experiments suggest the proposed method improves reconstruction of thin-structure objects.

**Compliance With Llm Reviewing Policy:**

Affirmed.

**Final Justification:**

All my concerns have been well addressed, and I will maintain my positive score.

**Key Questions For Authors:**

See Weaknesses.

**Limitations:**

No. The current evaluation appears largely focused on object-level or controlled settings, so it remains unclear how well MPFs generalize to complex real-world scans (e.g., scene-level data with diverse thin structures and clutter). In addition, MPFs relies on enforcing constraints on points assumed to lie on (or very near) the zero level set; under noisy or off-surface samples, this assumption may be violated and could degrade reconstruction quality.

**Strengths And Weaknesses:**

## Strengths

1. The proposed MPFs is accompanied by relatively simple proofs and derivations.
2. Both metrics and visual results indicate notable improvements on objects with thin structures.

## Weaknesses

1. Could the paper provide separate visualizations of the metric field and the phase/sign field to better illustrate their respective contributions within MPFs?

2. The paper lacks quantitative analysis specifically targeting thin-structure regimes (e.g., varying the separation between two nearby surfaces) to evaluate the stability of UDF, SDF, and MPF, and to identify when/why the method fails.

3. Experiments are mainly on synthetic point clouds with many qualitative results and relatively limited quantitative comparisons. Providing results on commonly used benchmarks (e.g., [1]) would better demonstrate generalization.

4. On real point clouds (Table 2 and Figure 9), MPF’s metrics are close to StEik and DiGS, but the visual reconstructions appear markedly different. Please clarify why a small metric gap corresponds to a large qualitative difference.

5. Although the paper includes one scanned-object example, this alone does not establish generalization in real scenes. Thin structures also occur in street-scene scans; please provide corresponding comparisons on such real-world data.

6. When input point clouds are noisy, it is difficult to provide high-quality on-surface points for supervision. Since MPFs’ modeling/derivations enforce constraints on samples lying on (or very near) the zero level set, how sensitive is the method to noise-induced off-surface samples, and does this degrade reconstruction quality?

## References

[1] I-Filtering: Implicit Filtering for Learning Neural Distance Functions From 3D Point Clouds

---

> ### Author Rebuttal · Authors · 2026-03-31
>
> We thank the reviewer for the insightful comments. We have included additional quantitative comparisons and visualizations in the [Anonymous Rebuttal Link](https://anonymous.4open.science/r/ICML-rebuttal-20018/rebuttal.md), and we would like to clarify several points below.
>
> 1. Visualization of metric and phase fields.
>
> We agree that separate visualization of the metric and phase/sign fields would help better illustrate their respective roles. Such visualizations are included in Figure 3 (anonymous link), and we will provide clearer explanations in the revised manuscript.
>
> 2. Quantitative analysis on thin-structure regimes.
>
> We thank the reviewer for this valuable suggestion. We conducted experiments on watertight disk-shaped models with varying thickness (normalized T = 0.005, 0.01, 0.05, 0.1). Figure 9 (anonymous link) shows that MPF consistently reconstructs thin structures across all regimes. In contrast, SDF often fails for very thin disks because the network cannot accurately capture the sign transitions in narrow regions. UDF-based methods frequently produce broken surfaces or separate layers, as the lack of explicit sign information prevents enforcing a coherent surface. These results demonstrate that MPF handles thin-structure reconstruction more robustly across varying thicknesses.
>
> 3. Evaluation on Diverse Datasets.
>
> We appreciate the reviewer’s suggestion regarding broader evaluation and agree that I-Filtering is a strong and well-designed work. In response, we have conducted extensive additional experiments covering over 20 comparisons across diverse scenarios (see anonymous link):
> - Scene-level datasets (Figures 4 and 6) including large-scale indoor and outdoor scenes, testing the ability to reconstruct complex geometry and thin structures.
> - Indoor real scans (Figure 5, ScanNet RGB-D point maps), containing significant noise and missing regions, highlighting robustness under realistic sensor perturbations.
> - Partial real-scan datasets (Figure 8, SRB dataset), with missing regions, noise, and outliers, demonstrating accurate reconstructions despite these challenges.
> - LiDAR street-scene point clouds (Figure 7), representing extremely sparse and noisy inputs.
> - Models with added Gaussian noise (Figure 8 bottom rows), evaluating stability under controlled perturbations.
>
> While the anonymous link currently provides primarily qualitative comparisons, these results consistently demonstrate that MPF produces high-quality reconstructions across a wide range of challenging scenarios. Due to time constraints, extensive numerical evaluations on comprehensive benchmark datasets are not yet included. We will incorporate additional large-scale benchmarks and report quantitative metrics in the final revision to provide a more thorough and complete evaluation.
>
>
> 4. Evaluation on real point clouds.
>
> We thank the reviewer for pointing this out. For this real-scan example, no ground-truth mesh is available, so the reported metric is computed as the average distance from the input points to the reconstructed surface. Consequently, methods that produce surfaces close to the points, even if they contain visible artifacts such as bulges or irregular geometry, can achieve similar scores. This explains why small metric differences can correspond to visually significant differences. We will clarify this evaluation protocol in the figure caption accordingly.
>
> 5. Generalization to real-world scenes.
>
> We have conducted additional experiments on real-scan datasets. As shown in anonymous link Figure 8 (first four rows), MPF demonstrates robustness in the presence of noise and missing data. We also include examples with clear thin structures in larger scenes (see link Figure 6), where MPF successfully recovers fine structures.
> For large-scale street-scene LiDAR data (see link Figure 7), we observe that reconstruction remains challenging due to significant outliers and sparsity. In such cases, global neural field methods may struggle. We plan to explore patch-based strategies (e.g., similar to CAP-UDF) to improve scalability and robustness in future work.
>
> 6. Sensitivity to noise and off-surface samples.
>
> We evaluate MPF under diverse noise conditions and input scenarios. In anonymous link Figure 5, the input corresponds to indoor scene data. The first three rows represent higher noise levels, where the inputs are not mesh-sampled points but point maps extracted from RGB-D videos in ScanNet, which contain significant noise. The subsequent rows instead use points sampled from the official meshes as input, resulting in cleaner observations.
> In addition to scene-level data, we also conduct experiments on watertight models with added Gaussian noise (0.5\%) (link Figure 8, last three rows), where MPF demonstrates stable reconstruction performance.
> Overall, these results suggest that MPF is reasonably robust to noise and off-surface perturbations in these scenarios.

---

> > ### Author Rebuttal · Reviewer_ZxGX · 2026-04-03
> >
> > Thanks for the author's response. All my concerns are well addressed.

---

> > > ### Author Response · Authors · 2026-04-03
> > >
> > > Dear Reviewer ZxGX,
> > >
> > > We sincerely thank the reviewer for the positive feedback. We are glad that the additional experiments and analyses provided in the rebuttal have successfully addressed all your concerns.
> > >
> > > Following your valuable suggestions, we will further improve the manuscript in the final version by:
> > >
> > > - Incorporating the additional experiments presented in this rebuttal into the manuscripts, along with more comprehensive quantitative comparisons.
> > > - Including the quantitative analysis on thin-structure regimes presented in this rebuttal to better highlight the advantages of our method.
> > > - Clarifying the evaluation protocol for real-scan data in the figure captions.
> > >
> > > We truly appreciate your time and constructive feedback, which have helped us improve the quality and clarity of our work.
> > >
> > > Best regards,
> > > The Authors

---

### Official Review · Reviewer_ZPji · 2026-03-06

**Soundness:** 3
**Presentation:** 3
**Significance:** 3
**Originality:** 2
**Overall Recommendation:** 4
**Confidence:** 4

**Summary:**

This paper introduces an approach to disentangle the metric distance and the sign in SDF-based 3D reconstruction. The method aims to solve well-known problems on SDFs and UDFs, namely 1) SDFs create artifacts when the geometry of the object to reconstruct is not watertight, and 2) both SDFs and UDFs struggle with thin structures in 3D. The authors prove that the properties of their disentangled approaches are appropriate for implicit 3D reconstruction and align well with Eikonal-based methods.

**Compliance With Llm Reviewing Policy:**

Affirmed.

**Final Justification:**

The paper is well written and theoretically sound. However, I worry about the number of loss functions that seem to me can cause problems when generalizing this method to other data or more challenging scenarios. However, I keep my weak acceptance score.

**Key Questions For Authors:**

1. The authors claim that their method can help reconstruct thin structures such as fins, petals, etc. Can other methods just reconstruct these structures with more capacity, more fine grate supervision, and higher resolution reconstructions (Marching cubes)? How is your method better/faster than the alternative?

2. See Weakness (ii).

3. Could you explain further the sentence: "Modeling \theta(x) as a continuous field avoids the instability associated with discrete sign switching." (Line 152, right). To what discrete switching are you referring here?

4. Line 227 refers to a small constant \epsilon that does not seem to appear in Eq. 8. Additionally, the authors mention squaring the dot product, but this is not reflected in the equation.

5. In line 377, how sensitive is the model to the loss weights? since there are many losses. The authors only mention an stability strategy for L_align.

**Limitations:**

The paper does not discuss the limitations of the proposed method beyond comparison with the baselines. Since 3D reconstruction is a crowded subfield of computer vision, I encourage you to explicitly list what are the limitations of the author's method to inform the scientific community.

I do not see any direct negative societal impact arising from the presented work.

**Strengths And Weaknesses:**

### Strengths

1. The paper presents an interesting (Niche) problem in implicit 3D reconstruction, i.e., reconstruction of thin structures and non-water-tight geometries.
2. The proposed distanglement, although simple in nature, allows defining specialized loss functions to control their respective behaviour and regularize the learning process.
3. Propositions 3.1 and 3.2 prove that the properties of the author's distangle method resemble those of an SDF and UDF field.
4. Regarding the experiments, the separation into two groups (UDF-target and SDF-target) for comparison is meaningful and aids in the analysis of the presented method's properties.

### Weaknesses

1. The authors claim that
2. In Eq. 3, as you approach the shape's surface, the value of P(x) affects the metric distance since it weights down the value r(x), potentially creating strange changes in the distance field. I suppose you can control this with a bigger \beta, since it is learned, but none of the loss functions seem to regularize that.
3. While all the losses are justified, this method uses significantly more losses than other approaches, e.g., usually they only use a data loss, an Eikonal loss, and perhaps one or two other regularization losses. This approach uses 8 different losses. My concern is an overconstrained optimization that might make these losses fight with each other. If this is not the case, an analysis needs to be provided.

---

> ### Author Rebuttal · Authors · 2026-03-31
>
> We thank the reviewer for the insightful comments. We have included additional quantitative comparisons and visualizations in the [Anonymous Rebuttal Link](https://anonymous.4open.science/r/ICML-rebuttal-20018/rebuttal.md), and we would like to clarify several points below.
>
> 1. Effect of $\beta$ in Eq. 3.
>
> We thank the reviewer for pointing out this aspect. In our implementation, $\beta$ is initialized to 50 with a learning rate of 1e-4. As described in the appendix, this choice provides a good balance between convergence speed and numerical stability in practice.
> While $\beta$ is learnable, we observe stable training behavior across experiments without additional explicit regularization. We agree that more principled control of $\beta$ could further improve robustness, and we will investigate this direction in future work.
>
>
> 2. Multiple losses and potential over-constrained optimization.
>
> We agree that our method introduces more loss terms compared to standard SDF-based approaches. However, these losses are designed to be complementary rather than conflicting: since MPF decouples the metric and phase fields, each set of losses focuses on a separate objective—distance regularization for the metric field and sign/phase consistency for the phase field—so they do not interfere with each other.
> - Several losses (e.g., $L_{zero}$, $L_{eik}$, $L_{far}$, $L_{normal}$) are commonly used in SDF-based methods.
> - The core MPF losses ($L_{align}$, $L_{phase}$, $L_{lap}$) guide the learning process for stable and accurate reconstructions.
>
> We further conducted ablation studies (see link Table 2, Figure 1), where we either remove individual losses or increase their weights (×10). The results show that:
> - Removing key losses leads to noticeable degradation,
> - Increasing weights results in only minor changes,
>
> indicating that the optimization is not overly sensitive and that the losses work cooperatively. While our formulation introduces additional constraints, they are necessary to prevent degeneracies in thin-structure reconstruction.
>
> 3. Sensitivity to loss weights.
>
> As discussed above, our ablation experiments indicate that the method is relatively robust to the choice of loss weights. Most weight variations lead to limited performance changes, while removing specific losses has a more significant impact. We will include a clearer discussion of this in the revised manuscript.
>
>
> 4. Comparison with increasing capacity / resolution in SDF.
>
> We thank the reviewer for this important question. We conducted additional experiments (see link Table 3, Figure 2) by increasing network capacity, point density, and extraction resolution for SDF-based methods.
> We observe that under practical experimental settings, SDF still struggles to reliably reconstruct thin structures. Blindly increasing network depth may increase model capacity but can also lead to instability. Raising the Marching Cubes resolution can help if the SDF is correctly learned, but offers limited improvement when the SDF itself is not accurately estimated. Increasing input point density can be beneficial to some extent; however, under extremely dense sampling, the notion of “thin structures” becomes relative to the sampling resolution, effectively reducing the difficulty of the problem rather than fundamentally addressing it.
>
>
> 5. Continuous phase field vs. discrete sign switching.
>
> We clarify that the "discrete sign switching" refers to formulations where the sign is modeled via discontinuous decisions (e.g., binary classification or hard sign assignment). Such representations introduce non-differentiable transitions near the surface, which can lead to instability during optimization.
> In contrast, MPF models $\theta(x)$ as a continuous field and defines:
> $P(x) = \tanh(\beta \cdot \theta(x))$,
> which provides smooth transitions with non-zero gradients near the boundary. This continuous formulation offers stable gradient signals and improves convergence to accurate surface locations.
>
> 6. Minor issues (Eq. 8, notation, constants).
>
> We thank the reviewer for pointing out these inconsistencies. We will revise the text to remove the unnecessary $\epsilon$ term and clarify the formulation of the inner product to ensure consistency with the implemented loss.

---

> > ### Author Rebuttal · Reviewer_ZPji · 2026-04-03
> >
> > While I think the idea is interesting, I still have concerns about the number of losses. For instance, I disagree when the authors say that all the losses are complementary. For instance, the Laplacian loss and the surface data loss are well-known to fight each other, as the first tends to over-smooth the reconstructed surface while the latter pushes for more accurate surfaces, including high-frequency details (not smooth).
> >
> > Nevertheless, I keep my score.

---

> > > ### Author Response · Authors · 2026-04-03
> > >
> > > Dear Reviewer ZPji,
> > >
> > > Thank you for the insightful comment and for recognizing the interesting idea in our paper. We agree that, in general, the Laplacian regularization and the surface data term can be conflicting: the former encourages smoothness, while the latter enforces accurate reconstruction of potentially high-frequency surface details.
> > >
> > > In our formulation and implementation, we explicitly account for this issue through a region-specific design. The surface data term is enforced only on surface samples, whereas the Laplacian regularization is applied only to **sparsely sampled points** in the ambient space. As described in Eq. (12), we further exclude a neighborhood around the surface when applying the Laplacian term. This significantly reduces its influence near the interface and near thin structures, where preserving geometric accuracy is especially important.
> > >
> > > To further validate this design, we conducted an ablation study on the David model in which the Laplacian loss is applied to different spatial regions (see the table below). The results show that applying the Laplacian term near the surface indeed harms reconstruction quality, leading to oversmoothing and loss of details. Applying it over the entire space has a smaller negative effect, likely because the samples are sparse, but it still performs worse than our design. In contrast, restricting the Laplacian loss to the non-surface band (our default setting) achieves the best result. These observations suggest that, although the two losses serve different purposes (smoothness vs. fidelity), their conflict can be effectively mitigated through our spatially selective regularization strategy.
> > > | Laplacian-applied region           | Samples | Chamfer Distance ↓ |
> > > | ---------------------------------- | ------: | -----------------: |
> > > | Excluding near-surface band (ours) |     15k |              0.938 |
> > > | Entire space                       |     15k |              0.979 |
> > > | Near-surface band only             |     15k |              1.221 |
> > >
> > > Best regards,
> > >
> > > The Authors

---

### Official Review · Reviewer_piyY · 2026-03-14

**Soundness:** 2
**Presentation:** 2
**Significance:** 3
**Originality:** 2
**Overall Recommendation:** 4
**Confidence:** 4

**Summary:**

This paper proposes a decoupled field including the (unsigned) metric and the phase. When combined, it can achieve similar behavior as signed distance fields, thus allowing efficient marching cubes iso-surface extraction. The decoupled representation also supports  representing thin layers, similar to UDF.

**Compliance With Llm Reviewing Policy:**

Affirmed.

**Ethical Review Flag:**

Flag this paper for an ethics review.

**Final Justification:**

Although I raised quite a few concerns, the authors made a good job responding to these, and clarified that many of the issues are due to optimization difficulty. Therefore, I raised my score to Weak Accept.

**Key Questions For Authors:**

Why the proposed method can achieve better results compared with existing SDF, especially if the surface is closed.

If a thin layer is turned into a thin structure, would SDF be able to achieve similar results?

**Limitations:**

Limitations should be discussed more clearly.

**Strengths And Weaknesses:**

Strengths:
- The proposed representation achieves a unified approach to handling closed surface and thin structures.
- The paper shows competitive performance compared with existing methods.

Weaknesses:
- It is unclear why the method achieves better performance, e.g. for closed surfaces, as SDF can do a good job.
- It is unclear whether the separation of metric and phase leads to a network with more learnable weights.
- The idea of separating the metric and phase is relative straightforward and the method implementation is also fairly standard.
- The fact that thin structure leads to thin layers with opposite signs seems to imply this could be achieved using an SDF of a thin layer, so the benefit of the proposed method is not clear.

---

> ### Author Rebuttal · Authors · 2026-03-31
>
> We thank the reviewer for the thoughtful questions and suggestions. We have included additional quantitative comparisons and visualizations in the [Anonymous Rebuttal Link](https://anonymous.4open.science/r/ICML-rebuttal-20018/rebuttal.md), and we would like to clarify several points below.
>
> (1) For common watertight geometries (e.g., Bunny, Armadillo), SDF-based methods indeed perform well, and MPF achieves comparable results in these cases. Our goal is not to replace SDF for such standard shapes.
>
> (2) For watertight shapes containing thin structures (e.g., Spikeball, Ship, Artichoke in Figure 6), MPF produces more accurate and stable reconstructions, while SDF-based methods often fail under the same settings. As shown in Table 1 and Figure 9 (link), MPF consistently outperforms SDF across varying thickness regimes.
>
> (3) This difference arises from the learning process. In standard SDF learning, sign and distance are tightly coupled, requiring the network to predict both surface proximity (metric) and consistent inside–outside labeling. In thin or tightly enclosed regions, the correct sign must flip rapidly within a narrow interval, which this coupling makes hard to capture, often causing thickness inflation or surface merging. In contrast, MPF decouples metric and phase, allowing the phase field to handle sign transitions independently, leading to more accurate and stable reconstruction of thin structures.
>
> We address each question below.
> 1. Why does MPF outperform SDF on closed surfaces?
>
> The key issue is not representational expressiveness but optimization difficulty. In standard SDF learning, sign and distance are tightly coupled: the network must simultaneously encode correct inside/outside labeling and surface proximity at the same time. This tight coupling makes it difficult to capture rapid sign flips in thin shells or tightly enclosed cavities, often causing thickness inflation, surface merging, or spurious closures.
>
> MPFs decouple these two objectives. The metric field $r(x)$ focuses purely on capturing unsigned proximity (regularized via the Eikonal constraint on near-surface samples), while the phase field $\theta(x)$ independently handles the sign transition. Because the phase head is not constrained to maintain unit gradients everywhere, it can resolve fine-scale sign changes—even across extremely thin gaps—without conflicting with the distance regularizer. This makes the optimization landscape better-conditioned, yielding more faithful reconstructions on watertight surfaces with thin structure.
>
> 2. Does the separation lead to more learnable weights?
>
> As described in Section 3.5, we use a shared SIREN backbone with two lightweight heads (each 256→256→1). The total number of parameters is comparable to standard single-head SDF networks (e.g., DiGS, StEik). The performance gain comes from a more structured inductive bias—not from increased model capacity.
>
> 3. The idea is straightforward / implementation is standard.
>
> We appreciate this observation. Indeed, the formulation is deliberately simple—and we view this as a strength. The novelty lies not in architectural complexity but in identifying and resolving a fundamental tension in SDF learning. Despite its simplicity, the decoupling produces non-trivial consequences: (i) Propositions 3.1–3.2 establish that the zero level set and surface gradients are preserved; (ii) the residual phase injection (Eq. 3) provides a theoretically grounded mechanism for stable near-surface gradients; and (iii) the approach enables direct Marching Cubes extraction on geometries that defeat both SDF and UDF baselines, as demonstrated across diverse models in Tables 1–2, Figures 6–11 and additional results in link Figures 4–8.
>
> 4. SDF of a thin layer could achieve similar results.
>
> In principle, an SDF can represent a watertight thin shell. However, in practice, SDF-based methods often struggle with thin structures due to optimization difficulties. As shown in Figure 6 (e.g., Ship, Artichoke), MPF reconstructs these thin-shell models reliably, while SDF-based methods may fail under identical sampling conditions. This advantage is consistently supported by empirical results: as shown in Table 1 and Figure 9 (link), MPF outperforms SDF across varying thickness regimes. We attribute this improvement to the decoupling of metric and phase in MPF, which alleviates the sign–distance coupling issue in SDF and leads to more stable behavior in thin regions.
>
> As noted earlier, our goal is not to replace SDF for standard watertight geometries, but to address its limitations in thin-structure scenarios. In this context, MPF also naturally handles open boundaries by effectively encoding them as thin shells during optimization, enabling robust reconstruction for both open scans and watertight inputs without additional preprocessing.

---

> > ### Author Rebuttal · Reviewer_piyY · 2026-04-03
> >
> > I believe the rebuttal has addressed my concerns, so I will raise my score to Weak Accept.

---

> > > ### Author Response · Authors · 2026-04-04
> > >
> > > Dear Reviewer piyY,
> > >
> > > We sincerely thank you for the positive feedback and for confirming that our responses have addressed your concerns. We greatly appreciate your helpful suggestions. In the final revision, we will incorporate the following improvements:
> > >
> > > - Clarify the scope of MPF and strengthen the explanation of the core idea, highlighting the benefit of decoupling metric and phase.
> > > - Include additional quantitative and visual results from the rebuttal to further support our claims.
> > >
> > > Thank you again for your time and for the constructive feedback that has helped improve our paper.
> > >
> > > Best regards,
> > >
> > > The Authors

---

### Decision · Program_Chairs · 2026-04-30

**Decision:**

Accept (regular)

**Comment:**

All four reviewers converged on Weak Accept, and the rebuttal was effective in resolving the majority of concerns.
Reviewers consistently recognized the value of the core idea, empirical results on synthetic thin-shell and thin-plate shapes, and the additional thickness sensitivity analysis, field visualizations, scene-level and real-scan data, and the 1-Lipschitz backbone comparison.
Remaining concerns are minor but worth addressing in the final revision:  relationship to Points2Surf, Marching Cubes resolution, cleaner ablation in the main text. Limitations including sensitivity to noise, behavior on large-scale scenes, and the thin-shell approximation for open surfaces, should be discussed explicitly.